# Increasing ocean wave energy observed in Earth's seismic wavefield since the late 20th century

Richard C. Aster [1] ✉, Adam T. Ringler[2], Robert E. Anthony [2] & Thomas A. Lee[3]

Ocean waves excite continuous globally observable seismic signals. We use data from 52 globally distributed seismographs to analyze the vertical component primary microseism wavefield at 14–20 s period between the late 1980s and August 2022. This signal is principally composed of Rayleigh waves generated by ocean wave seafloor tractions at less than several hundred meters depth, and is thus a proxy for near-coastal swell activity. Here we show that increasing seismic amplitudes at $3\sigma$ significance occur at 41 (79%) and negative trends occur at $3\sigma$ significance at eight (15%) sites. The greatest absolute increase occurs for the Antarctic Peninsula with respective acceleration amplitude and energy trends ($\pm 3\sigma$) of $0.037 \pm 0.008$ nm s$^{-2}$y$^{-1}$ ($0.36 \pm 0.08\%$ y$^{-1}$) and $4.16 \pm 1.07$ nm$^2$ s$^{-2}$y$^{-1}$ ($0.58 \pm 0.15\%$ y$^{-1}$), where percentage trends are relative to historical medians. The inferred global mean near-coastal ocean wave energy increase rate is $0.27 \pm 0.03\%$ y$^{-1}$ for all data and is $0.35 \pm 0.04\%$ y$^{-1}$ since 1 January 2000. Strongly correlated seismic amplitude station histories occur to beyond 50° of separation and show regional-to-global associations with El Niño and La Niña events.

Earth's seismic wavefield was revealed by the late 19th century to be incessantly excited at periods between ~8 and 30 s. It was well established by the 1960s that seafloor forces due to wind-driven ocean gravity waves are the principal source of seismic waves in this period range in the absence of earthquakes or other large transient events, and that this microseismic wavefield is primarily composed of seismic surface waves[1]. Mid-20th century studies established phenomenological understanding of the distinct ocean wave to solid Earth source coupling processes responsible for the primary ($\approx$14–20 s) and secondary ($\approx$6–12 s) microseism period bands. Microseism signals occur at much longer periods than typical anthropogenic seismic noise[2] and thus constitute an ocean-state proxy that is well recorded by seismographs essentially everywhere on Earth.

Global microseism signals are generated by two distinct source processes. The primary microseism between ~14 and 20 s is excited by normal and shear tractions due to dynamic pressure variations of ocean swell on the near-coastal seafloor[3–5]. Ocean waves in this period range have wavelengths $\lambda$ in the range of 300–600 m, and their dynamic pressure field decays exponentially with depth at a rate determined by the wavenumber-depth product. The primary microseism spectrum[6] matches the swell spectrum of partially to fully developed seas excited by worldwide storm systems, and microseism amplitudes reflect near-coastal seafloor tractions influenced by swell amplitude, propagation direction, and/or wave period. The distinct and more energetic secondary microseism between ~6 and 12 s, not analyzed in this study, has been investigated more thoroughly than the primary mechanism. It arises from wave period-shifted pressure perturbations on the ocean floor generated by generalized standing (clapotic) components of the ocean wavefield resulting from swell interference. These standing wave components may be generated by wave interactions occurring within a single translating storm system, between swell arising at multiple storm centers, or between incident

[1]Department of Geosciences, Warner College of Natural Resources, Colorado State University, 801 S. Howes St., Fort Collins 80523-1482 CO, USA. [2]Albuquerque Seismological Laboratory, U.S. Geological Survey, Target Rd. 10002 Isleta SE, Kirtland AFB, Albuquerque 87117 NM, USA. [3]Department of Earth and Planetary Sciences, Harvard University, 20 Oxford St., Cambridge 02138 MA, USA. ✉e-mail: rick.aster@colostate.edu

and reflected waves due to coasts[7]. Notable efforts in the theory and numerical modeling of seismic ocean source processes in recent years have sought to more thoroughly incorporate necessary elements of seafloor bathymetry, physical oceanography, and seismology[7–11] to refine the understanding of the ocean swell wavefield responsible for the secondary microseism and the influences of water depth, wave propagation direction, seafloor slope, and general bathymetry on primary and secondary microseism generation[12,13].

Increasing ocean basin surface wind speeds since the mid-20th century have been inferred from meteorologic, oceanographic, and satellite altimetry data. The greatest wave powers and power increases are attributed to the Southern Ocean between 40° and 80° S[14,15]. Relevant factors vary geographically over multi-decade timescales under the influence of troposphere and ocean warming[16–21] with[14] estimating that global wave power has on average increased by $1.087 \times 10^3$ kW m$^{-1}$ y$^{-1}$ and correlates strongly with sea surface temperature for 1948–2008. The microseism wavefield arises from geographically distributed forces applied to the seafloor by wind-driven ocean wave activity and is thus a proxy for the ocean wave state that complements surface and remote measurements[6]. This indicates that secular and other trends in ocean waves state are expected to be globally reflected in seismic data. Microseism-based wave state studies spanning shorter timescales than presented here have identified spatiotemporal trends in storm intensity, duration, and tracking[22–26], and have been applied to estimate high-latitude sea ice variations and associated ocean wave attenuation[27,28]. Wave state is modulated by inter-annual climate processes[14,29–31] and extreme microseism intensity associations, particularly with El Niño (ENSO) states, have previously been noted using shorter duration data sets[25]. However, global evidence for widespread secular intensification of microseism amplitudes has not been previously documented.

In this work, we assess primary microseism intensity since the late 20th century using data from globally distributed research and monitoring seismographs. We demonstrate a strong prevalence of increasing seismic amplitudes and energies at rates that are consistent with ocean wave intensification estimates from other disciplines. We additionally perform a correlation and clustering analysis of seismic amplitude time series to demonstrate that the primary microseism signal is a consistent proxy for long-range ocean wave spatial and temporal variability.

## Results

### Calculation of microseism metrics

We calculated microseism metrics from power spectral densities (PSDs) through 1 August 2022 using the section averaging method applied for the conterminous United States in ref. 32. The PSD estimation methodology is similar to that of ref. 33 but does not implement smoothing or binning beyond the inherent frequency discretization and spectral leakage properties of the discrete Fourier transform. We calculate seismic spectra in 1-hour 50% overlapping windows from continuous long-period high-gain vertical component (channel LHZ) 1 sample per second seismic data, which is insensitive to Rayleigh wave propagation direction, recorded by primary station sensors (location code 00) retrieved from the EarthScope Data Management Center (DMC) from the IU[34] and II[35] networks. Additional detail on spectral estimation is included in Methods.

Global earthquakes occur with a clustered Poissonian time distribution[36] and large shallow events produce strong seismic surface waves that overlap with the 14–20 s primary microseism period band[37,38]. We remove potential earthquake transients from global earthquakes with magnitudes ≥5.75 using origin times and magnitudes from the U.S. Geological Survey Comprehensive Catalog of Earthquake Events and Products (ComCat)[39] to cull intervals during which the primary microseism may be obscured. ComCat event selection is based on W phase[40] moment magnitude $M_{ww}$ or on body wave magnitude $m_b$ for smaller earthquakes, and we remove data for all stations

beginning at the earthquake origin time for 3, 6, 12, 24, and 48 hours for magnitude ranges 5.75 ≤ M ≤ 7.0, 7.0 ≤ M ≤ 7.5, 7.5 ≤ M ≤ 8.0, 8.0 ≤ M ≤ 9.0, and M ≥ 9, respectively. These data removal windows are highly conservative given that the global transit time for surface waves in this period range is about 3 hours and only very large earthquakes produce significant multi-circumglobal signals[37].

Seismic data contain calibration pulses that produce values that greatly exceed naturally occurring levels, and may also contain periods of nonphysically low or high amplitude values corresponding to local physical disturbances or technical malfunctions. We remove these nonphysical outliers by comparing integrated signal power to the global New High-/Low-Noise models (NHNM/NLNM) of ref. 41, applying an acceptance corridor for the primary microseism period band between five times that is predicted by the (only rarely reached) low-noise model and 50 times that are predicted by the high-noise model.

### Characterization of stationary seasonal variations

Seismic data commonly exhibit large annual variability in microseism intensity corresponding to the winter development of extratropical cyclonic storms, high-latitude formation of sea ice, and other seasonal factors[24,27,28]. We characterize stationary annual variations as a standard Fourier series representation $H(t)$ in which the Fourier coefficients for the tropical year fundamental and its first three harmonic coefficients are estimated by projecting microseism amplitude $A(t)$ or energy (calculated from the square of seismic velocity) $E(t)$ time series onto orthogonal trigonometric basis functions at periods $T_0$, $T_0/2$, $T_0/3$, and $T_0/4$ to obtain corresponding coefficients $a_i$ and $b_i$

$$H_{A,E}(t) = a_0 \sin(t/T_0) + b_0 \cos(t/T_0) + a_1 \sin(2t/T_0) + b_1 \cos(2t/T_0) \\ + a_2 \sin(3t/T_0) + b_2 \cos(3t/T_0) + a_3 \sin(4t/T_0) + b_3 \cos(4t/T_0) \quad (1)$$

where $T_0 = 365.242$ days. Aperiodic sampling and data gap tolerant Lomb-Scargale PSD analysis[42] of daily sampled median acceleration amplitude time series and examination of decaying coefficient ($a_i$, $b_i$) amplitudes confirmed that four Fourier terms were sufficient to characterize all significant spectral lines for our purposes.

### Robust trend estimation

To assess the influences of removing ComCat catalog earthquake windows, noise model culling, and the presence or subtraction of station-specific stationary annual harmonic (equation (1)) functions, we applied robust ($\ell_1$-norm minimizing; see Methods) trend estimation to progressively processed data sets for each station. The removal of short data intervals using (1) earthquake catalog and (2) noise model determined outlier procedures, as described above, produces trend estimates with lesser uncertainties but does not appreciably affect overall assessment of global seismic amplitude and energy trends nor the conclusions of this study (e.g., Supplementary Fig. 1).

Hourly time series as described above are smoothed using a two-month (61-day) moving median in daily steps to produce 1 sample per day series signals for $\ell_1$-norm minimizing linear trend determination. We obtain trend estimates for signals within the primary microseism band (14–20 s) at global broadband seismic stations selected solely for operational histories exceeding 20 years and data completeness exceeding 80%. These selection criteria allow for the estimation of primary microseism trends at 52 globally distributed stations (Table 1) with earliest data ranging from 1988 to 1999 (Supplementary Fig. 2).

We fit seismic vertical acceleration amplitudes $A(t)$ (Fig. 1) with $\ell_1$-norm minimizing linear functions to estimate annualized rates of amplitude change $R_A$ (Fig. 2a, b; Table 2) for each station. Seismic wave energy is proportional to the square of velocity amplitude, and we estimate these time series $E(t)$ (Supplementary Fig. 3) to obtain annualized rates of change, $R_E$ (Fig. 2c; Table 3). Corresponding proportional annual percentage change rates $P_A$ and $P_E$ are calculated by

**Table 1 | Station names, sites, International Federation of Digital Seismograph Networks (FDSN) network codes, locations, and creation dates**

| Station name | Site name | Net. Code | Latitude | Longitude | Elevation | Creation date |
|---|---|---|---|---|---|---|
| AAK | Ala Archa, Kyrgyzstan | II | 42.6375 | 74.4942 | 1633.1 | 10-12-1990 |
| ANMO | Albuquerque, New Mexico, USA | IU | 34.9459 | −106.4572 | 1850.0 | 08-29-1989 |
| BFO | Black Forest Observatory, Schiltach, Germany | II | 48.3301 | 8.3296 | 638.0 | 05-29-1996 |
| BORG | Borgarfjordur, Asbjarnarstadir, Iceland | II | 64.7474 | −21.3268 | 110.0 | 07-30-1994 |
| CASY | Casey, Antarctica | IU | −66.2792 | 110.5354 | 10.0 | 02-19-1996 |
| CHTO | Chiang Mai, Thailand | IU | 18.8141 | 98.9443 | 420.0 | 08-31-1992 |
| COLA | College Outpost, Alaska, USA | IU | 64.873599 | −147.8616 | 200.0 | 06-14-1996 |
| COR | Corvallis, Oregon, USA | IU | 44.5855 | −123.3046 | 110.0 | 10-26-1989 |
| CTAO | Charters Towers, Australia | IU | −20.0882 | 146.2545 | 357.0 | 06-17-1991 |
| DWPF | Disney Wilderness Preserve, Florida, USA | IU | 28.1103 | −81.4327 | 30.0 | 08-02-1998 |
| EFI | Mount Kent, East Falkland Island | II | −51.6753 | −58.0637 | 110.0 | 02-16-1996 |
| ESK | Eskdalemuir, Scotland, UK | II | 55.3167 | −3.205 | 242.0 | 11-13-1987 |
| FFC | Flin Flon, Canada | II | 54.725 | −101.9783 | 338.0 | 08-28-1993 |
| GUMO | Guam, Mariana Islands | IU | 13.5893 | 144.8684 | 170.0 | 06-09-1991 |
| HKT | Hockley, Texas | IU | 29.9618 | −95.8384 | -413.0 | 07-11-1995 |
| HRV | Adam Dziewonski Observatory (Oak Ridge), Massachusetts, USA | IU | 42.5064 | −71.5583 | 200.0 | 09-22-2008 |
| INCN | Inchon, Republic of Korea | IU | 37.4776 | 126.6239 | 80.0 | 07-20-1995 |
| KBS | Ny-Alesund, Spitzbergen, Norway | IU | 78.9154 | 11.9385 | 90.0 | 11-05-1994 |
| KDAK | Kodiak Island, Alaska, USA | II | 57.7828 | −152.5835 | 152.0 | 06-09-1997 |
| KEV | Kevo, Finland | IU | 69.7565 | 27.0035 | 100.0 | 06-07-1993 |
| KIP | Kipapa, Hawaii, USA | IU | 21.42 | −158.0112 | 110.0 | 08-15-1988 |
| KIV | Kislovodsk, Russia | II | 43.9562 | 42.6888 | 1210.0 | 09-14-1988 |
| KONO | Kongsberg, Norway | IU | 59.6491 | 9.5982 | 216.0 | 06-20-1991 |
| KURK | Kurchatov, Kazakhstan | II | 50.7154 | 78.6202 | 184.0 | 03-26-1995 |
| LCO | Las Campanas Astronomical Observatory, Chile | IU | -29.011 | −70.7005 | 2274.0 | 08-04-2014 |
| LVZ | Lovozero, Russia | II | 67.8979 | 34.6514 | 630.0 | 12-01-1992 |
| MAJO | Matsushiro, Japan | IU | 36.54567 | 138.20406 | 405.0 | 08-18-1990 |
| MDJ | Mudanjiang, Heilongjiang Province, China | IC | 44.617 | 129.5908 | 270.0 | 11-09-1996 |
| NNA | Nana, Peru | II | −11.9875 | −76.8422 | 575.0 | 06-22-1988 |
| OBN | Obninsk, Russia | II | 55.1146 | 36.5674 | 160.0 | 09-14-1988 |
| PAB | San Pablo, Spain | IU | 39.5446 | −4.3499 | 950.0 | 10-20-1992 |
| PAYG | Puerto Ayora, Galapagos Islands | IU | -0.6742 | −90.2861 | 270.0 | 06-19-1998 |
| PET | Petropavlovsk, Russia | IU | 53.0233 | 158.6499 | 110.0 | 08-28-1993 |
| PFO | Pinon Flat, California, USA | II | 33.6092 | −116.4553 | 1280.0 | 10-24-1986 |
| PMG | Port Moresby, New Guinea | IU | −9.4047 | 147.1597 | 90.0 | 09-10-1993 |
| PMSA | Palmer Station, Antarctica | IU | −64.7744 | −64.0489 | 40.0 | 03-03-1993 |
| PTCN | Pitcairn Island, South Pacific | IU | −25.0713 | −130.0953 | 220.0 | 12-29-1996 |
| RAR | Rarotonga, Cook Islands | IU | −21.2125 | −159.7733 | 28.0 | 03-07-1992 |
| RSSD | Black Hills, South Dakota, USA | IU | 44.1212 | −104.0359 | 2090.0 | 09-24-1999 |
| SBA | Scott Base, Antarctica | IU | −77.8492 | 166.7572 | 50.0 | 10-28-1998 |
| SHEL | Horse Pasture, St. Helena Island | II | −15.9594 | −5.7455 | 537.0 | 06-19-1995 |
| SJG | San Juan, Puerto Rico | IU | 18.1091 | −66.15 | 420.0 | 05-26-1993 |
| SNZO | South Karori, New Zealand | IU | −41.3087 | 174.7043 | 120.0 | 04-07-1992 |
| SSPA | Standing Stone, Pennsylvania | IU | 40.6358 | −77.8876 | 270.0 | 12-01-1994 |
| SUR | Sutherland, South Africa | II | −32.3797 | 20.8117 | 1770.0 | 10-30-1990 |
| TATO | Taipei, Taiwan | IU | 24.9735 | 121.4971 | 160.0 | 09-26-1992 |
| TAU | Hobart, Tasmania, Australia | II | −42.9099 | 147.3204 | 132.0 | 01-17-1994 |
| TSUM | Tsumeb, Namibia | IU | −19.2022 | 17.5838 | 1260.0 | 08-19-1994 |
| TUC | Tucson, Arizona | IU | 32.3098 | −110.7847 | 910.0 | 06-13-1992 |
| ULN | Ulaanbaatar, Mongolia | IU | 47.8651 | 107.0532 | 1610.0 | 10-31-1994 |
| WRAB | Tennant Creek, NT, Australia | II | −19.9336 | 134.36 | 366.0 | 03-27-1994 |
| YAK | Yakutsk, Russia | IU | 62.031 | 129.6805 | 110.0 | 08-31-1993 |

Primary sensors are installed in observatory vaults or boreholes[65] with associated metadata maintained by the U.S. Geological Survey, EarthScope Data Management Center, and the International Federation of Digital Seismograph Networks.

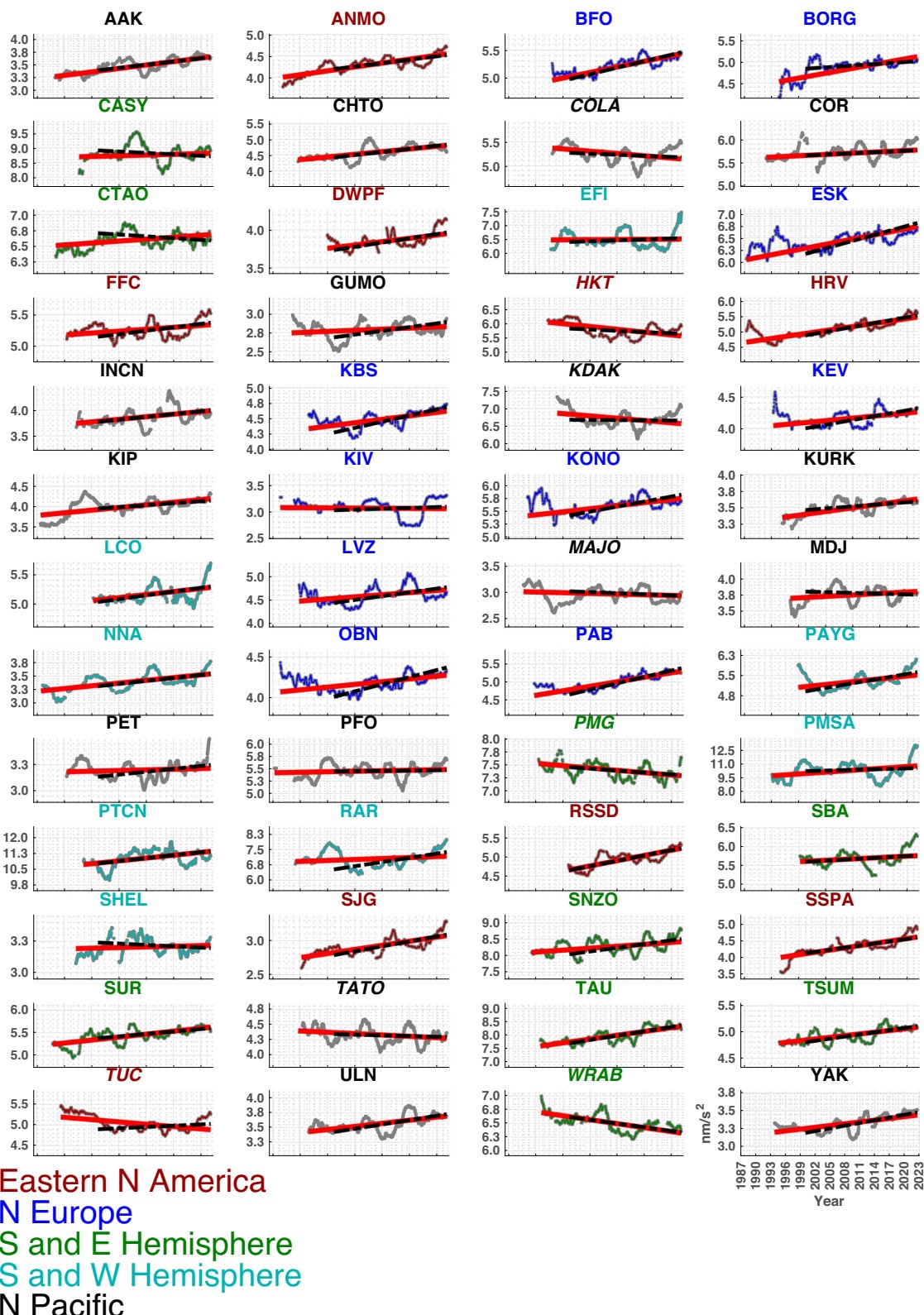

**Fig. 1 | Primary microseism vertical component acceleration amplitude histories at 52 long-operational seismic stations with associated robust trend estimates.** Trends (equation (6)) are estimated for time series with stationary seasonal harmonic functions (equation (1)) subtracted (Fig. 2, Table 2). Time series are displayed after applying 3-year moving median data smoothing for plotting clarity while trend values are calculated from daily sampled two-month (61-day) moving median filtered data. Trends for all available data, and for post-2000 data, are shown in red and black, respectively. Title colors indicate latitude, longitude

(ϕ, ℓ) defined regions as follows: Blue: European North Atlantic (ϕ > 0°, −43° < ℓ < 43°); Red: Mid-North America and North Atlantic (0° < ϕ < 55°, −111° < ℓ < −60°); Cyan: Southwest Hemisphere (ϕ < 0°, ℓ < 0°); Green: Southeast Hemisphere (ϕ < 0°, ℓ > 0°); Black: Northern Hemisphere Pacific and Asia outside of Blue and Red groups. Time series with negative trends have italicized titles. Time axis tick marks correspond to 1 January of the indicated years. Corresponding seismic energy histories are shown in Supplementary Fig. 3.

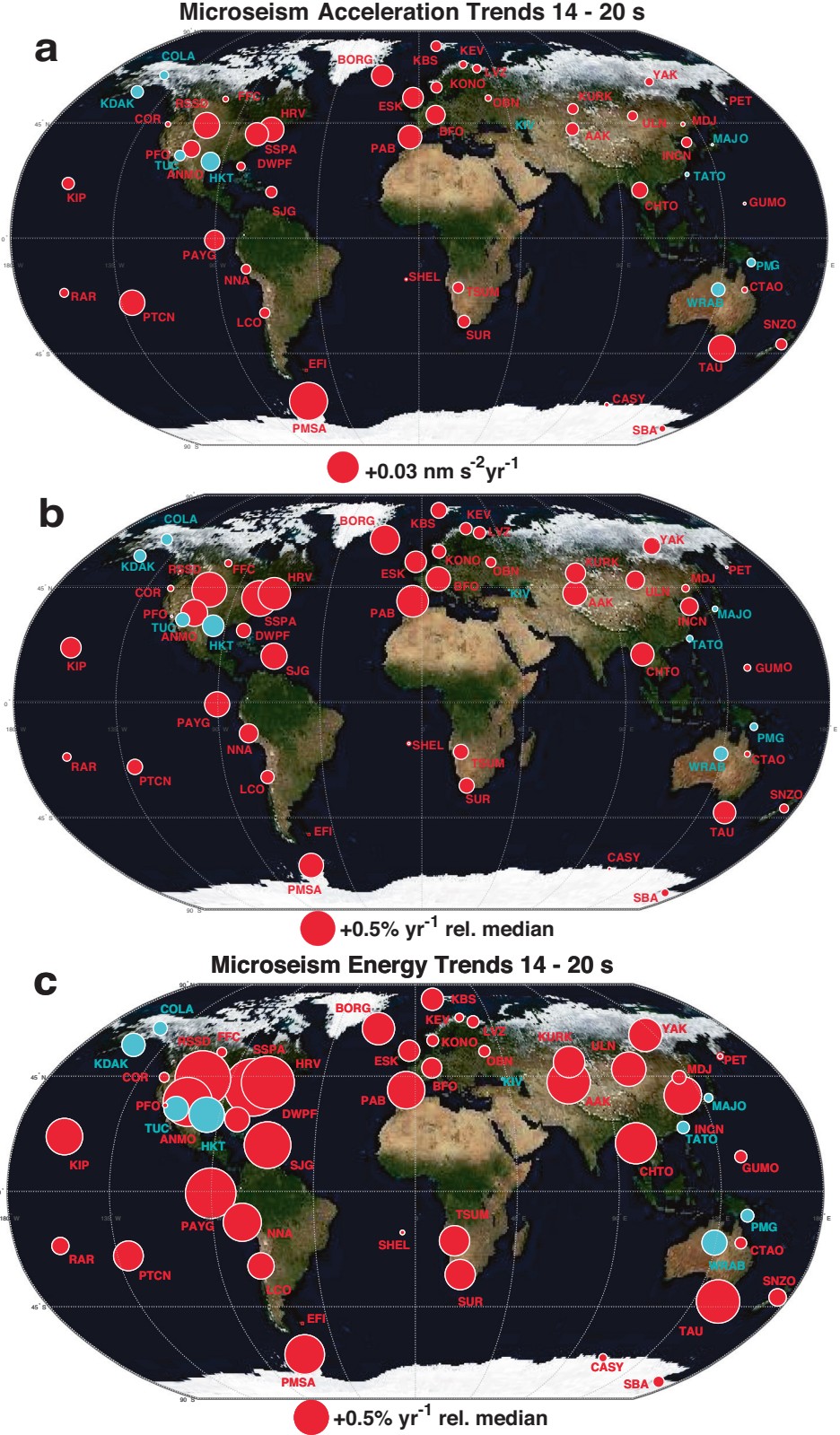

**Fig. 2 | Station locations and global trends (red positive, cyan negative) for vertical component acceleration amplitude, vertical component acceleration amplitude normalized by historical median, and vertical component seismic energy normalized by historical median. a** absolute ($R_A$), **b** percentage ($P_A$; equation (2); Table 2) seismic acceleration amplitude trends (red: positive; cyan: negative). **c** Percentage velocity squared energy proxy trends ($P_E$; equation (3); Table 3). Circle radius is proportional to the trend value and white rims indicate $3\sigma$ trend significance (Fig. 3).

**Table 2 | Acceleration secular trends (equation (2)) with stationary annual harmonic functions (equation (1)) removed, sorted by amplitude (Figs. 1, 2)**

| Station | $R_A$ (nm s$^{-2}$ y$^{-1}$) | $3\sigma$ (nm s$^{-2}$ y$^{-1}$) | $P_A$ (%y$^{-1}$) | $3\sigma$ (%y$^{-1}$) |
|---|---|---|---|---|
| PMSA | 0.037 | 0.008 | 0.358 | 0.080 |
| TAU | 0.026 | 0.003 | 0.326 | 0.033 |
| RSSD | 0.025 | 0.002 | 0.509 | 0.049 |
| PTCN | 0.025 | 0.007 | 0.224 | 0.060 |
| HRV | 0.024 | 0.002 | 0.471 | 0.031 |
| PAB | 0.023 | 0.002 | 0.461 | 0.048 |
| SSPA | 0.022 | 0.002 | 0.520 | 0.046 |
| BORG | 0.021 | 0.003 | 0.425 | 0.067 |
| ESK | 0.020 | 0.003 | 0.311 | 0.045 |
| PAYG | 0.019 | 0.003 | 0.366 | 0.057 |
| BFO | 0.019 | 0.003 | 0.355 | 0.060 |
| ANMO | 0.017 | 0.001 | 0.390 | 0.025 |
| CHTO | 0.015 | 0.002 | 0.338 | 0.040 |
| AAK | 0.012 | 0.001 | 0.358 | 0.040 |
| SUR | 0.012 | 0.002 | 0.221 | 0.029 |
| KIP | 0.012 | 0.001 | 0.298 | 0.033 |
| SJG | 0.011 | 0.001 | 0.380 | 0.041 |
| KONO | 0.011 | 0.003 | 0.196 | 0.055 |
| TSUM | 0.011 | 0.002 | 0.219 | 0.032 |
| SNZO | 0.011 | 0.003 | 0.131 | 0.032 |
| KBS | 0.010 | 0.002 | 0.230 | 0.050 |
| KURK | 0.010 | 0.002 | 0.293 | 0.048 |
| LCO | 0.010 | 0.003 | 0.193 | 0.057 |
| INCN | 0.010 | 0.002 | 0.258 | 0.048 |
| ULN | 0.010 | 0.001 | 0.270 | 0.039 |
| NNA | 0.009 | 0.001 | 0.276 | 0.029 |
| RAR | 0.009 | 0.003 | 0.123 | 0.047 |
| LVZ | 0.008 | 0.003 | 0.180 | 0.059 |
| YAK | 0.008 | 0.001 | 0.250 | 0.031 |
| DWPF | 0.008 | 0.002 | 0.210 | 0.044 |
| KEV | 0.007 | 0.002 | 0.178 | 0.057 |
| SBA | 0.007 | 0.003 | 0.116 | 0.057 |
| OBN | 0.006 | 0.002 | 0.147 | 0.042 |
| CTAO | 0.006 | 0.002 | 0.086 | 0.025 |
| FFC | 0.006 | 0.002 | 0.106 | 0.035 |
| COR | 0.005 | 0.002 | 0.092 | 0.039 |
| CASY | 0.004 | 0.004 | 0.051 | 0.051 |
| MDJ | 0.004 | 0.002 | 0.112 | 0.041 |
| GUMO | 0.003 | 0.001 | 0.097 | 0.049 |
| PFO | 0.002 | 0.002 | 0.035 | 0.031 |
| PET | 0.001 | 0.001 | 0.036 | 0.034 |
| EFI | 0.001 | 0.004 | 0.016 | 0.064 |
| SHEL | 0.001 | 0.001 | 0.029 | 0.033 |
| KIV | −0.001 | 0.002 | −0.019 | 0.054 |
| MAJO | −0.002 | 0.001 | −0.082 | 0.042 |
| TATO | −0.004 | 0.002 | −0.094 | 0.040 |
| PMG | −0.008 | 0.002 | −0.111 | 0.029 |
| COLA | −0.009 | 0.002 | −0.166 | 0.046 |
| TUC | v0.010 | 0.002 | −0.205 | 0.031 |
| KDAK | −0.012 | 0.003 | −0.178 | 0.050 |
| WRAB | −0.013 | 0.002 | −0.203 | 0.029 |
| HKT | −0.018 | 0.002 | −0.313 | 0.042 |

**Table 3 | Velocity squared energy (equation (3)) secular trends with stationary annual harmonic functions (equation (1)) removed, sorted by amplitude**

| Station | $R_E$ (nm$^2$ s$^{-2}$ y$^{-1}$) | $3\sigma$ (nm$^2$s$^{-2}$ y$^{-1}$) | $P_E$ (%y$^{-1}$) | $3\sigma$ (%y$^{-1}$) |
|---|---|---|---|---|
| PMSA | 4.157 | 1.071 | 0.576 | 0.148 |
| PTCN | 3.651 | 1.009 | 0.433 | 0.120 |
| TAU | 2.718 | 0.282 | 0.632 | 0.066 |
| HRV | 1.498 | 0.110 | 0.771 | 0.056 |
| RSSD | 1.473 | 0.160 | 0.810 | 0.088 |
| PAYG | 1.351 | 0.215 | 0.735 | 0.117 |
| SNZO | 1.181 | 0.296 | 0.252 | 0.063 |
| SSPA | 1.174 | 0.127 | 0.837 | 0.090 |
| ESK | 1.140 | 0.179 | 0.307 | 0.048 |
| PAB | 1.080 | 0.132 | 0.544 | 0.066 |
| BORG | 0.957 | 0.196 | 0.470 | 0.096 |
| ANMO | 0.922 | 0.065 | 0.714 | 0.050 |
| SUR | 0.888 | 0.116 | 0.440 | 0.057 |
| CHTO | 0.868 | 0.117 | 0.599 | 0.080 |
| RAR | 0.846 | 0.324 | 0.250 | 0.095 |
| TSUM | 0.735 | 0.108 | 0.441 | 0.065 |
| LCO | 0.696 | 0.205 | 0.383 | 0.113 |
| BFO | 0.687 | 0.149 | 0.292 | 0.063 |
| CASY | 0.677 | 0.475 | 0.120 | 0.084 |
| KIP | 0.594 | 0.072 | 0.531 | 0.064 |
| INCN | 0.561 | 0.108 | 0.545 | 0.105 |
| AAK | 0.537 | 0.066 | 0.630 | 0.078 |
| KBS | 0.522 | 0.131 | 0.322 | 0.081 |
| KONO | 0.495 | 0.165 | 0.170 | 0.057 |
| CTAO | 0.492 | 0.147 | 0.168 | 0.050 |
| ULN | 0.449 | 0.070 | 0.500 | 0.078 |
| NNA | 0.427 | 0.045 | 0.555 | 0.058 |
| KURK | 0.413 | 0.084 | 0.458 | 0.093 |
| SJG | 0.412 | 0.049 | 0.678 | 0.080 |
| DWPF | 0.392 | 0.092 | 0.361 | 0.085 |
| SBA | 0.387 | 0.229 | 0.165 | 0.098 |
| YAK | 0.378 | 0.047 | 0.484 | 0.060 |
| COR | 0.348 | 0.157 | 0.143 | 0.065 |
| LVZ | 0.309 | 0.144 | 0.174 | 0.081 |
| FFC | 0.291 | 0.135 | 0.140 | 0.065 |
| OBN | 0.244 | 0.072 | 0.164 | 0.048 |
| MDJ | 0.196 | 0.079 | 0.202 | 0.081 |
| KEV | 0.187 | 0.100 | 0.125 | 0.067 |
| PFO | 0.154 | 0.126 | 0.072 | 0.059 |
| GUMO | 0.099 | 0.072 | 0.177 | 0.129 |
| PET | 0.049 | 0.049 | 0.065 | 0.065 |
| SHEL | 0.047 | 0.046 | 0.067 | 0.066 |
| EFI | 0.035 | 0.355 | 0.013 | 0.127 |
| KIV | −0.027 | 0.063 | −0.036 | 0.084 |
| MAJO | −0.077 | 0.051 | −0.125 | 0.083 |
| TATO | −0.227 | 0.101 | −0.179 | 0.079 |
| COLA | −0.417 | 0.162 | −0.204 | 0.079 |
| TUC | −0.649 | 0.109 | −0.359 | 0.060 |
| PMG | −0.686 | 0.214 | −0.186 | 0.058 |
| WRAB | −1.056 | 0.169 | −0.365 | 0.059 |
| KDAK | −1.106 | 0.300 | −0.337 | 0.091 |
| HKT | −1.271 | 0.195 | −0.514 | 0.079 |

normalizing rates by corresponding station median amplitudes (Supplementary Fig. 4).

$$P_A = R_A / \text{med}(A(t)) \times 100\% \qquad (2)$$

$$P_E = R_E / \text{med}(E(t)) \times 100\% . \qquad (3)$$

For both acceleration $A(t)$ and velocity squared $E(t)$ trend results (seasonal harmonic signals removed), 41 stations (79%) show positive and 8 (15%) show negative slopes at $3\sigma$ significance (Figs. 2, 3; Tables 2, 3). Trend estimates obtained after subtraction of the seasonal harmonic signal (equation (1)) had lesser uncertainties reflecting the reduced signal mean average deviation from the linear function fit, but showed similar results to those obtained directly from $A(t)$ and $E(t)$ (Fig. 3, Supplementary Fig. 1).

Stations with significant positive amplitude and energy trends have wide geographically distribution and show regional magnitude correlations that are particularly well resolved for northeastern North America, western Europe (historically the most densely instrumented regions), and for sites in the extratropical southern hemisphere. Stations with significant negative trends are restricted to the northern and western Pacific Ocean regions and to two sites (TUC, Tucson, Arizona, and HKT, Hockley, Texas) in the southern United States. Seismic acceleration and velocity squared trends ($R_A$, $R_E$; Tables 2, 3) with associated $3\sigma$ uncertainties range from $(0.037 \pm 0.008 \, \text{nms}^{-2} \, \text{y}^{-1}$; $4.157 \pm 1.071 \, \text{nms}^{-2} \, \text{y}^{-1})$ at PMSA (Palmer Station, Antarctic Peninsula) to $(-0.018 \pm 0.002 \, \text{nms}^{-2} \, \text{y}^{-1}$; $-1.271 \pm 0.195 \, \text{nms}^{-2} \, \text{y}^{-1})$ at HKT. Trends exhibit a modest (correlation coefficient of 0.213) proportionality in that stations with high historical median primary microseism amplitudes (Supplementary Fig. 4) tend to also exhibit greater amplitude and energy increases (Fig. 4), as inferred for wave energy increase in the Southern Ocean[43].

The highest proportional rates $P_E$ of microseism energy increase are observed at eastern North America station SSPA (Standing Stone, Pennsylvania) and at central North America station RSSD (Black Hills, South Dakota) at $0.837 \pm 0.090$ and $0.810 \pm 0.088\% \, \text{y}^{-1}$, respectively (Table 3). The highest absolute rate of amplitude and energy increase is observed at PMSA (Palmer Station, Antarctica), but its corresponding proportional rate of energy increase ($0.576 \pm 0.148\% \, \text{y}^{-1}$) is not as high as at other (e.g., eastern North America) stations (Table 3), reflecting very high historical median primary microseism levels and perhaps the influence of variable Antarctic sea ice[28]. Similarly relatively moderate proportionate rates of energy increase are found for other high-energy southern hemisphere stations (TAU; Hobart, Tasmania, and PTCN; Pitcairn Island) which are expected to have sensitivity to the state of the Southern Ocean (Fig. 4).

## Discussion

Southern US stations TUC (Tuscon, Arizona) and HKT (Hockley, Texas), which were the most negative trending stations for the complete data interval, both change significantly (with TUC becoming positive at $3\sigma$ significance for acceleration and seismic energy) when the analysis is restricted to post 1 January 2000 data (Supplementary Fig. 5). Examining the time series (Fig. 1, Supplementary Fig. 3) for these two sites this is seen to arise from long-duration (>5 y) amplitude decreases between ~1993 and 2004 that subsequently reverse. These sites have some sensitivity to wave conditions in the Gulf of Mexico where overall downward general wave trends outside of hurricane season have been inferred for the earlier interval[44]. However, the overall microseism history of TUC and HKT indicates that they have the greatest sensitivity to the eastern North Atlantic wave state and they increasingly correlate with other such associated North American stations after 2005 (station cluster SAIP as described below). High absolute microseism amplitude increase rates for southern stations

such as TAU (Hobart, Tasmania), PMSA (Palmer Station, Antarctica), and PTCN (Pitcairn Island) are sustained in these later data and are consistent with studies showing increasing surface winds in the far southern hemisphere[14, 20]. Negative primary microseism amplitude trends in parts of the north and west Pacific region generally continue in the post-2000 era. This may reflect decadal-scale reanalysis and in situ observation supported conclusions of decreasing wave height in this region since the mid-1990s attributed to the strengthening of the negative phase of the Pacific-North American teleconnection between about 1996 and 2012 (e.g., as seen in microseism acceleration and seismic energy history at COLA (College, Alaska)) (Fig. 1)[45].

To examine a uniform interval during which all stations were in operation (Supplementary Fig. 2) we estimated secular trends solely for data recorded on or after 1 January 2000 (Fig. 1, Supplementary Figs. 3, 5; trend estimates for this shorter interval are also shown in Fig. 1). This nearly 23-year data set produces greater trend uncertainties but also exhibits overwhelmingly positive rates (e.g., 40 of 52 stations with positive acceleration trends with $3\sigma$ significance).

Cluster and global station stacks of amplitude and energy time series reveal correlated multi-year variations in near-coastal wave energy over the past 34 years in the primary microseism. We demonstrate this with a zero-lag correlation-based dendrogram analysis using detrended and demeaned 61-day-median smoothed vertical component acceleration time series with stationary annual harmonic functions (equation (1)) subtracted. We also incorporated identically smoothed Southern Oscillation Index (SOI) and El Niño Southern Oscillation (ENSO) indices (which are strongly anti-correlated; $c = -0.95$; Supplementary Figs. 6, 7) in this clustering. This clustering identifies geographically correlated primary microseism signals reflecting historical large-scale integrated ocean wave state. Southwestern Pacific sites (cluster SWP) correlate with the ENSO time series and Southeastern Pacific and Southwestern Atlantic (cluster SEPSWA) sites cluster with the (ENSO approximate additive inverse) SOI time series (Fig. 5, Supplementary Fig. 7). These associations are consistent with the[14] who noted correlations of 1948–2008 global wave power with the Niño3 standardized index in which increased wave energy occurs in the western equatorial Pacific and southeastern Pacific during positive and negative index excursions, respectively. We similarly tested for associations with North Atlantic Oscillation (NAO), Pacific Decadal Oscillation (PDO), Western Pacific Oscillation (WPO), and Atlantic Multidecadal Oscillation (AMO) time series and did not find comparably strong correlations for these indices with microseism amplitudes.

Secular and transient coherent changes in microseism amplitude and energy are visible on monthly and longer scales within regional clusters, and in some cases globally (Figs. 6, 7a). In particular, we note the large-scale influence of both El Niño and La Niña phases of ENSO, e.g., as visible in the closest associated respective station clusters (SWP and SEPSWA) and reflecting their dominant role in tropical Pacific climate variability[29] and influence on extreme storm frequency in the southwest Pacific region[30]. ENSO influences on global wave power are globally evident across this distribution seismographic stations as higher energy intervals correlating with positive and negative ENSO excursions spanning 2006–2007 (El Niño), 2010–2011 (La Niña), 2015–2016 (El Niño), and 2020 (La Niña) (Fig. 7b). The ENSO cyclicity shown in Fig. 7b is more globally widespread in the sense that it remains apparent in the median energy time series when the 11 SWP and SEPSWA stations are removed from the median seismic energy calculation (Supp. Fig. 8).

The strong 1997–2000 El Niño/La Niña ($A$ in Fig. 7), which is associated with a particularly prominent wave power peak in[14] is apparent but is less temporally distinct than for later excursions. This may be due to lower data completeness during this time (Supplementary Fig. 2). Figure 7b also displays global secular primary microseism energy trend estimates for all data and for the post 1 January 2000 epoch ($0.27\% \, \text{y}^{-1}$ and $0.35\% \, \text{y}^{-1}$, respectively; Fig. 7b). As noted above, microseism energy trends are somewhat lower than for the

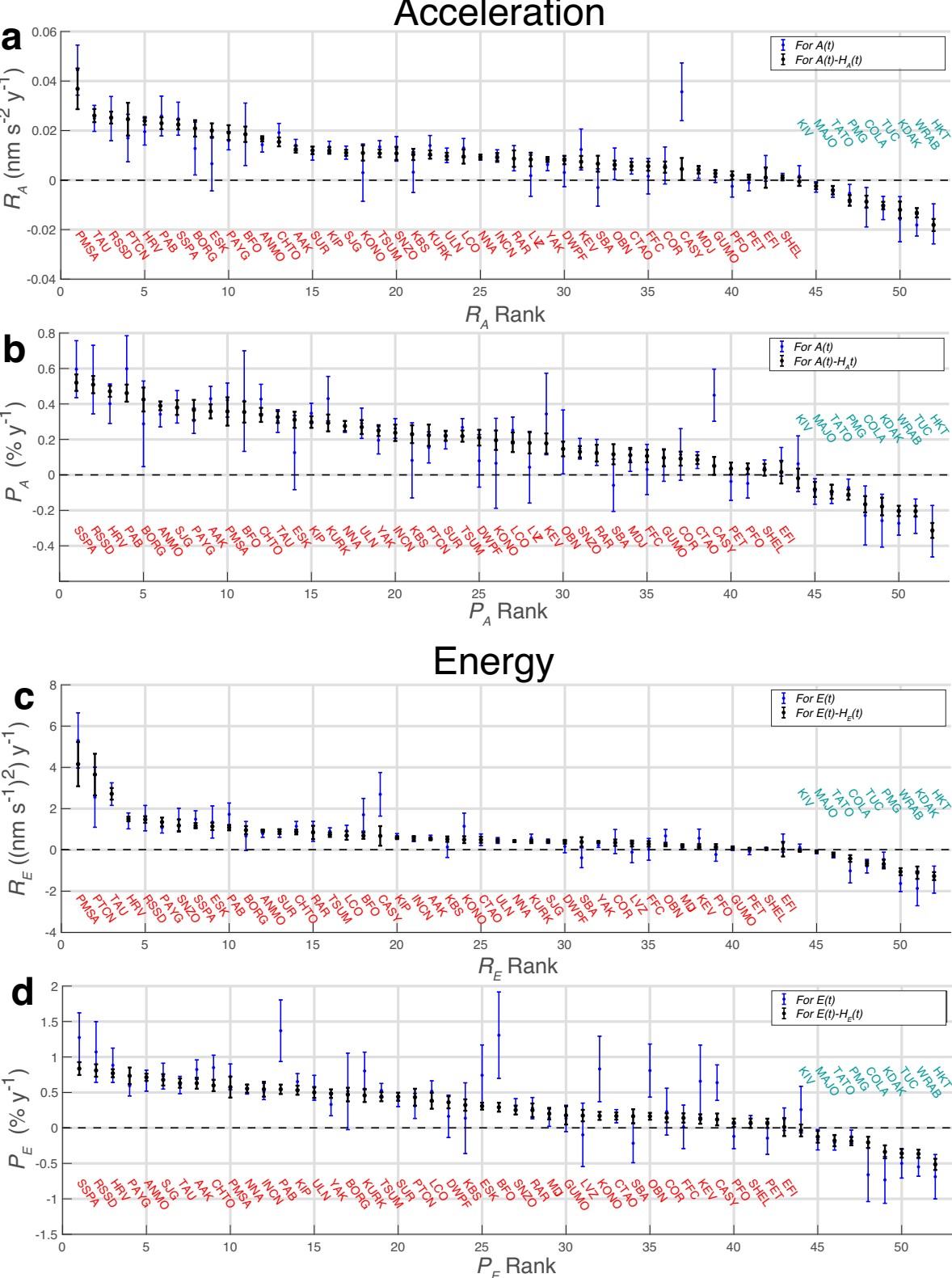

**Fig. 3 | Microseism trend results by station (red: positive; cyan: negative) sorted from most postive to most negative. a, b** Acceleration amplitude percentage trends (Fig. 2b; Table 2; equation (2)) for absolute and percentage trends, $R_A$ and $P_A$, respectively. **c, d** Energy (velocity amplitude squared) absolute and percentage trends $R_E$ and $P_E$ (Fig. 2c; Table 3; equation (3)). Blue and black data points and accompanying $3\sigma$ error bars reflect estimates obtained using the complete time series $A(t)$ and $E(t)$ and those obtained with associated seasonal harmonic trends (equation (1)) subtracted as indicated in subfigure legends. $x$ axis rank indicates the largest-to-smallest order of trends fitted to data with stationary seasonal components subtracted (black data points).

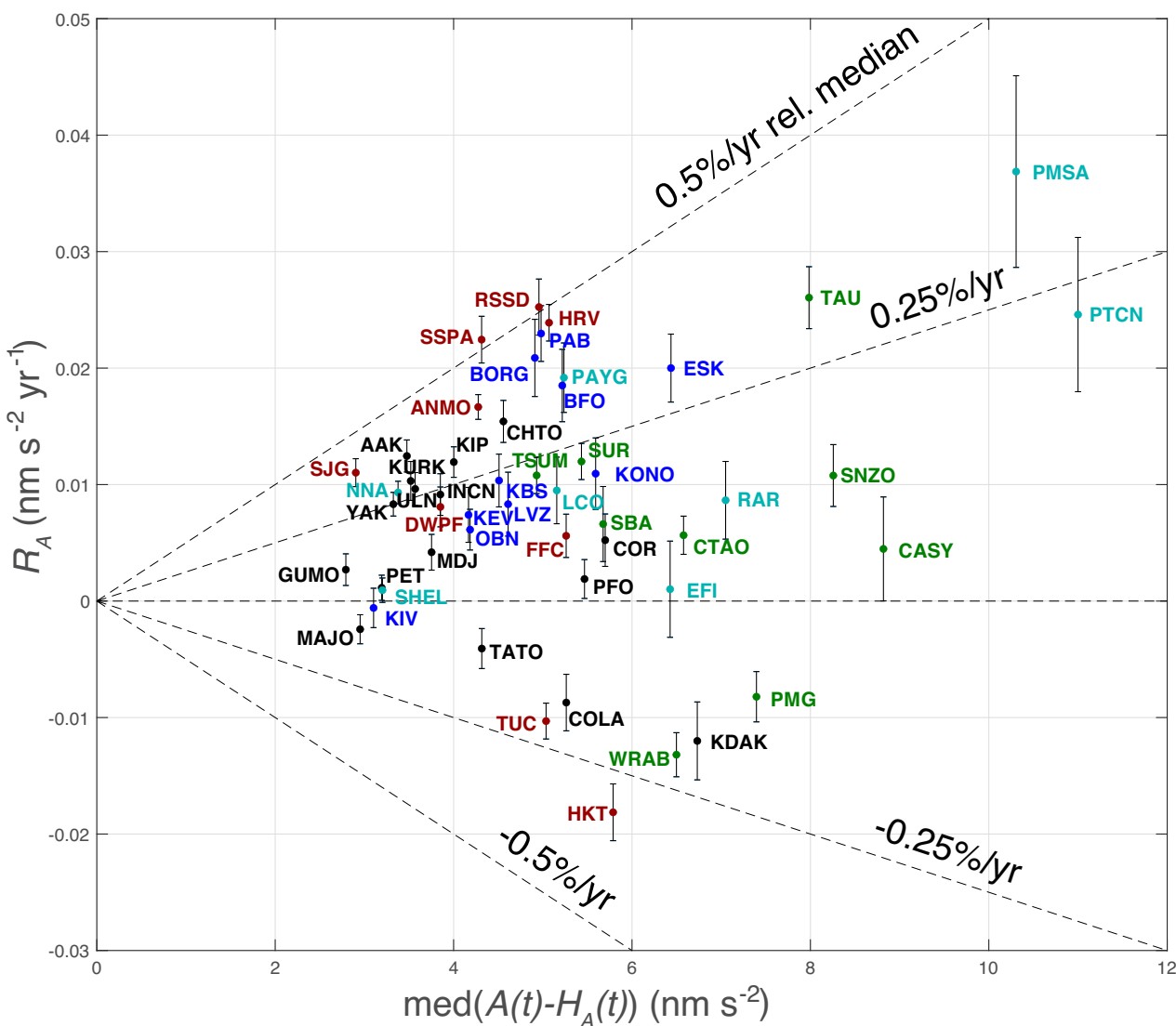

**Fig. 4 | Vertical component acceleration amplitude trends $R_A$ calculated with seasonal harmonics subtracted versus historical station median acceleration amplitude with $3\sigma$ confidence intervals.** The correlation coefficient is 0.213 and 13 stations exhibit positive trends at $3\sigma$ significance that are greater in absolute value than at the most negative station (HKT; Hockley, Texas). Colors reflect geographic groups defined in Fig. 1. Dotted lines indicate representative percentage amplitude changes $P_A$ relative to the historical station median (Figs. 1b, 2b, 4; Table 2).

1948–2008 all-oceans global trend estimated by[14]. This may reflect the relative brevity of our time series and/or the near-exclusive sensitivity of the primary microseism to near-coastal wave state.

The large correlation distance of inter-station primary microseism amplitudes, previously noted in ref. 4, is apparent in the acceleration time series correlation versus inter-station distance relationship (Fig. 7c), which also shows a slope change near a distance of 50°. We hypothesize that this slope break may represent a transition between sub-ocean basin-scale correlated storminess and swell teleconnection operating across large coastal expanses to ~50°, and to global- or near-global scale annual to multi-annual correlated wave intensities at greater distances (Fig. 7a, b). This correlation length scale indicates that primary microseism amplitudes and energies provide consistent proxies for large spatial and temporal scale ocean wave variability.

Wavelengths for primary microseism causative 14–20 s period deep-water waves are given by[46]

$$\lambda = \frac{gT^2}{2\pi} \qquad (4)$$

where $g$ is gravitational acceleration, and are thus approximately $\lambda = 310$–$620$ m. The dynamic pressure of idealized linearized (Airy) water waves decays with depth $z$ as

$$p(z) = e^{-\frac{2\pi z}{\lambda}}. \qquad (5)$$

Primary microseism-generating tractions thus attenuate to $p(z) = 0.05$ by $z/\lambda \approx 0.48$, and ocean regions for which the 14–20 s period primary microseism source mechanism is active are wavelength-dependent and correspondingly restricted to coastal and continental shelf regions with $z < 150$–$295$ m. This constitutes ~11% of the global seafloor lying almost exclusively along continental and island coasts ([47]; Fig. 5).

Spatio-temporally oscillatory tractions across variable bathymetry integrate to non-zero long-spatial wavelength seismic source terms, with the strongest coupling occurring at geographically limited (Fig. 5) regions of shallow seafloor and where bathymetric slopes are large[4]. Primary microseism amplitudes at a given site reflect a sensitivity kernel that depends on causative swell amplitude and direction,

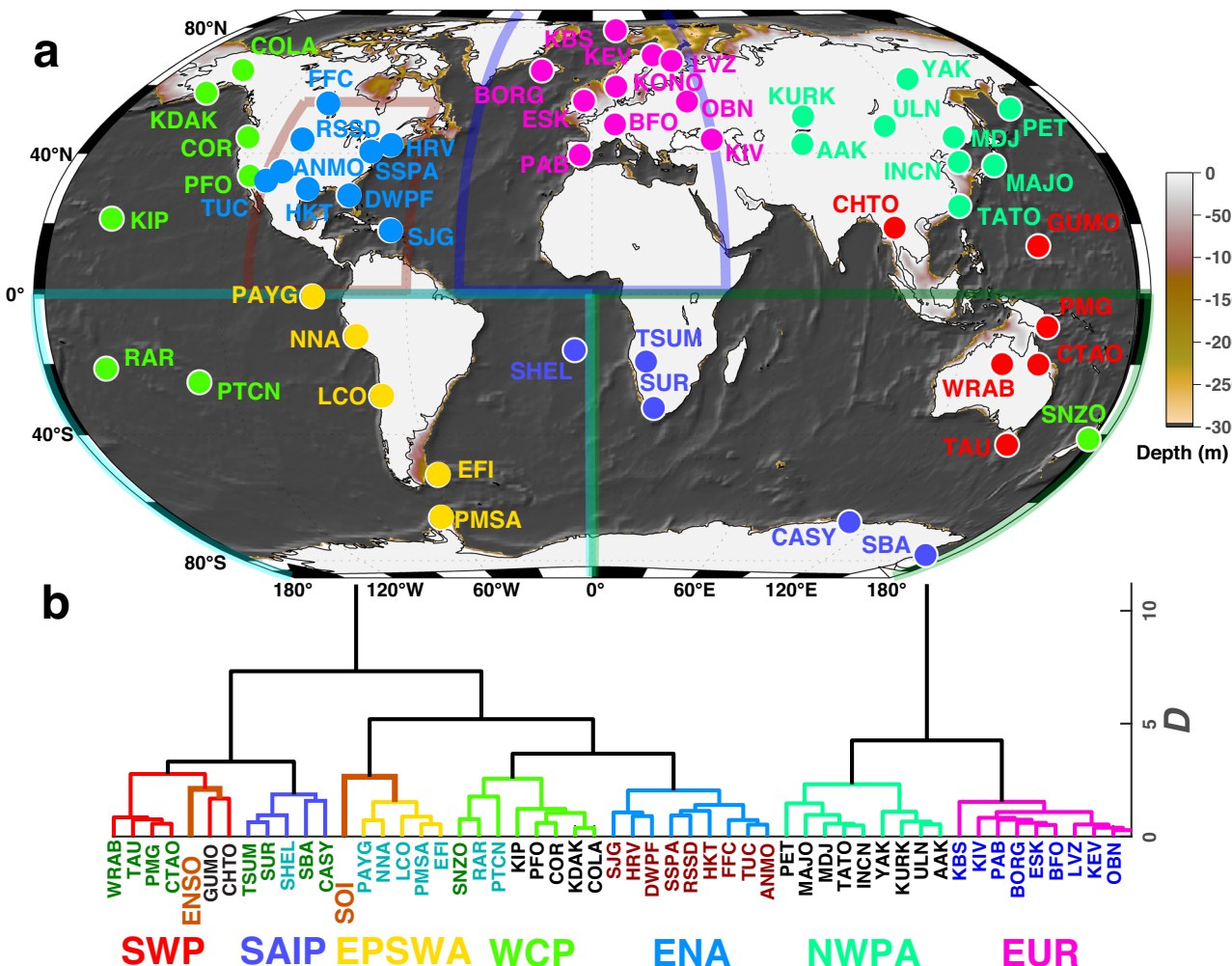

**Fig. 5 | Clustering results for all stations based on correlated vertical component seismic acceleration histories. a** Microseism acceleration station clustering derived from correlation (Fig. 6, Supplementary Fig. 6) of detrended 61-day moving median microseism acceleration amplitude time series with seasonal harmonics (equation (1)) and secular trends (Fig. 1; Table 2) removed. $D$ denotes the Ward dissimilarity metric[62]. Regions of ocean bathymetry with depths of <300 m and candidate primary microseism source zones (equation (5)) correspond to the color bar. Transparent geographic boundaries and their colors, and station names below the dendrogram, correspond to general regional associations noted in Fig. 1 and Supplementary Fig. 3. **b** Dendrogram classification of station groups corresponding to **a**: SWP Southwest Pacific, SAIP South Atlantic, Indian, Pacific, SEPSWA Southeast Pacific and Southwest Atlantic, WCP Western and Central Pacific, ENA Eastern North America, NWPEA Northwest Pacific and east Asia, EUR Europe and Southwest Asia. ENSO and SOI indicate dendrogram correlation-based associations for equivalently smoothed El Niño and Southern Oscillation index time series (Supplementary Fig. 7).

near-coastal bathymetry, seismic source efficiency, and wave period (equation (5)). Seismic surface wave propagation factors from near-coastal source zones to stations include $r^{-1}$ geometric spreading, seismic attenuation, and focusing or defocusing due to heterogeneous Earth structures. Complete modeling of these processes is a frontier effort in seismology, and studies for specific stations have shown qualified success[13,48]. Seismic polarization (i.e., incorporating horizontal seismic components) and array methods have been shown to image primary and secondary source regions at regional to global scales[4,10,48] and offer further impetus for improved modeling and model validation. The rate of median microseism amplitude and energy increase is a fraction of a percent per year and long-period global-scale seismic wave propagation characteristics are temporally invariant. Given these conditions, we suggest that ocean wave and seismic energy in these observations should be proportional, and specifically that changes in median primary microseism energy at the 61-day averaged scale as examined here will be proportional to similar time scale and geographically integrated changes in median ocean wave energy expressed as seafloor tractions across Earth's coastal regions.

Primary microseism observations at a variety of spatial and temporal scales constitute a period-dependent and complementary near-coastal-sensitive metric to multi-decade sea-state data collected at fixed ocean buoys[15,49] and inferences from carefully processed satellite altimeter data. Data integration and joint assessments of both ocean-wide and coast-proximate sea-state are expected to be increasingly fruitful as data quality, density, and analytical methodologies improve for multiple types of observation.

Global seismic records of the primary microseism resolve increasing near-coastal median energy in the ocean wavefield across a multi-decade time scale and long-range correlations and clustering between median-normalized signals indicate that these signals reflect long-range ocean wave state. Assuming uniform and linear coupling between ocean and seismic wavefield energy, the global average inferred integrated wave energy $P_E$ and corresponding seafloor tractions increase across 52 long-operational seismograph sites is $0.27 \pm 0.03\%$ y$^{-1}$ for the entire historical data set beginning in the late 1980s and $0.35 \pm 0.04\%$ y$^{-1}$ for post 1 January 2000 data. These estimates are geographically biased towards the high $P_E$ North Atlantic

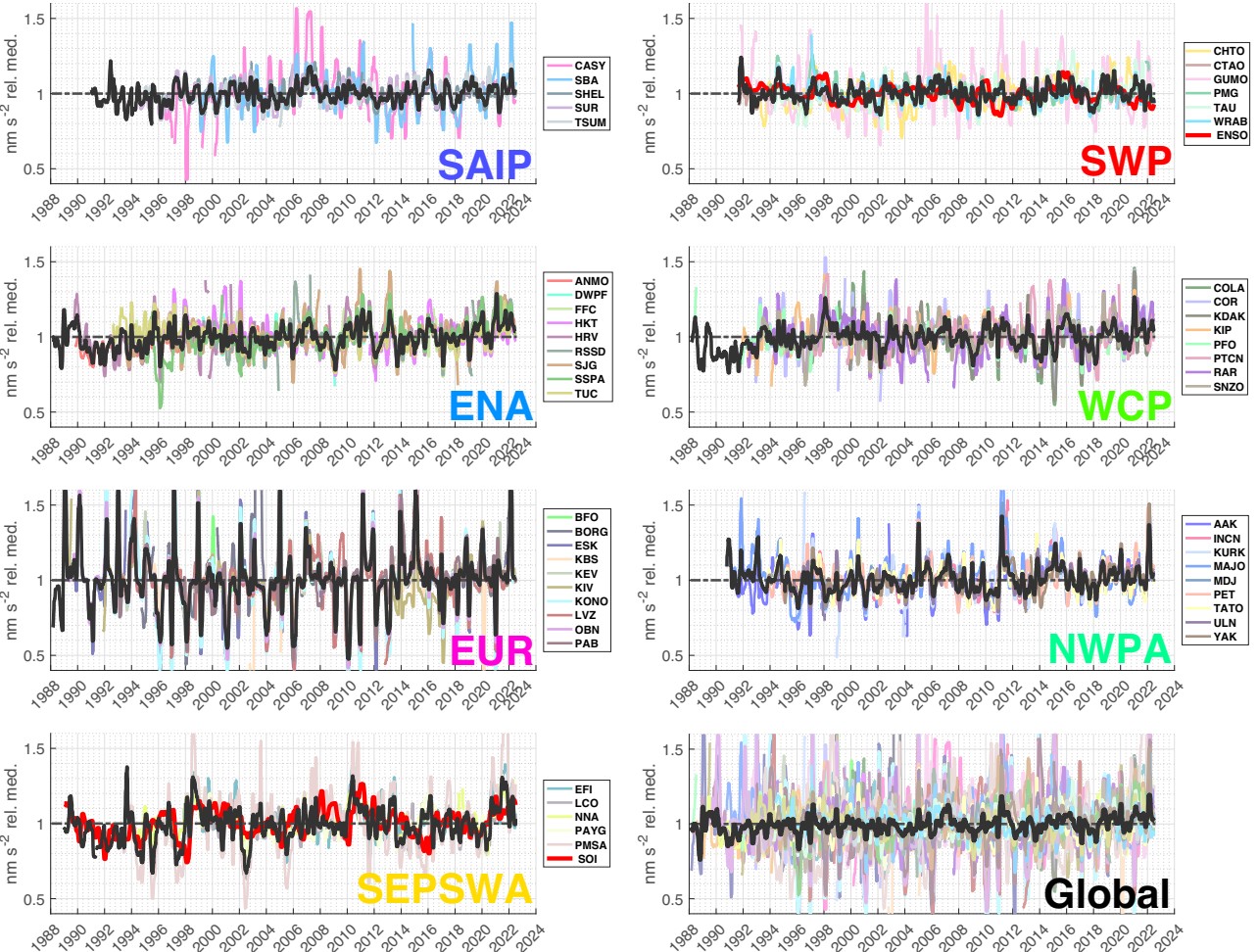

**Fig. 6 | Overlain vertical component acceleration time series (61-day smoothing) for station clusters and all stations.** Microseism acceleration time series clusters are defined from the associations shown in Fig. 5 and are normalized by respective station medians (Supplementary Fig. 4). Figure 7 shows corresponding 3-year smoothed time series. Black time series show the median of all smoothed time series for each subfigure. ENSO and SOI indices (Supplementary Fig. 7) scaled by seismic data are plotted in red within associated SWP and SEPSWA clusters, respectively.

region by historical seismographic station density (Fig. 2). This estimate of multidecadal global wave energy increase rate is comparable to that estimated in recent independent meteorological and oceanographic studies, e.g., the estimate of 0.4% y[-1] for recent secular rate of wave energy increase in[14]. The somewhat lower rates of change estimated here relative to the all-ocean analysis of ref. 14 (Fig. 7c) may reflect the near-coastal sensitivity of the primary microseism proxy relative to estimates from methodologies characterizing wave energy across entire ocean basins.

Monitoring and understanding changes in global and global near-coastal ocean wave state is central to projecting wave impacts, including as aggravated by sea level rise on coastal ecosystems, structures, and processes both natural and anthropogenic[50]. In this context, the primary microseism is a unique and swell period-sensitive metric for assessing wave-induced tractions that perform elastic and inelastic work on the shallow seafloor as well as for characterizing large-scale ocean wave state. Seismic data are freely distributed and telemetered to global seismological data centers for earthquake monitoring, tsunami warning, and other rapid-response missions, and microseism metrics and modeling can therefore be jointly interpreted with other data sources either retrospectively or in near-real time.

Expanding this analysis described in this study into the pre-1980s analog era of seismic recording prevalent throughout the 20th century offers an opportunity to quantitatively analyze microseism trends on

longer-term timescales, although absolute calibration in the pre-digital era presents challenges and identification of extreme storm events through statistical analysis of microseism amplitudes may be more fruitful for extracting information from many earlier data sets[25]. Analog instruments were highly sensitive to microseism signals (Supplementary Fig. 9) but early computational technology preempted long-term frequency-domain analyses. Steps are being taken towards the preservation of these historical archives and the development of robust and scaled digitization software to obtain centennial-scale microseism and other seismic metrics[51]. Notable studies in this regard are[52], where historical seismograms from the Royal Observatory of Belgium were digitized and used to study a historical storm event in 1953 and ref. 53, who extracted a 90-year wave height record for central California using data from the University of California, Berkeley.

## Methods
### Spectral estimation
Hourly PSD estimates were calculated via Welch's section averaging method using eleven 1024-s subwindows with 75% overlap. Additional stations with shorter contiguous recording intervals from the Federation of Digital Seismograph Networks (FDSN) networks IC[54], G[55], SR[56], AS[57], DW[58], and CU[59] were identically examined and display similar trends but are not included in this study due to shorter operational intervals. Each estimate requires at least 90% data completeness within each 1-hour window. We zero-pad any data gaps

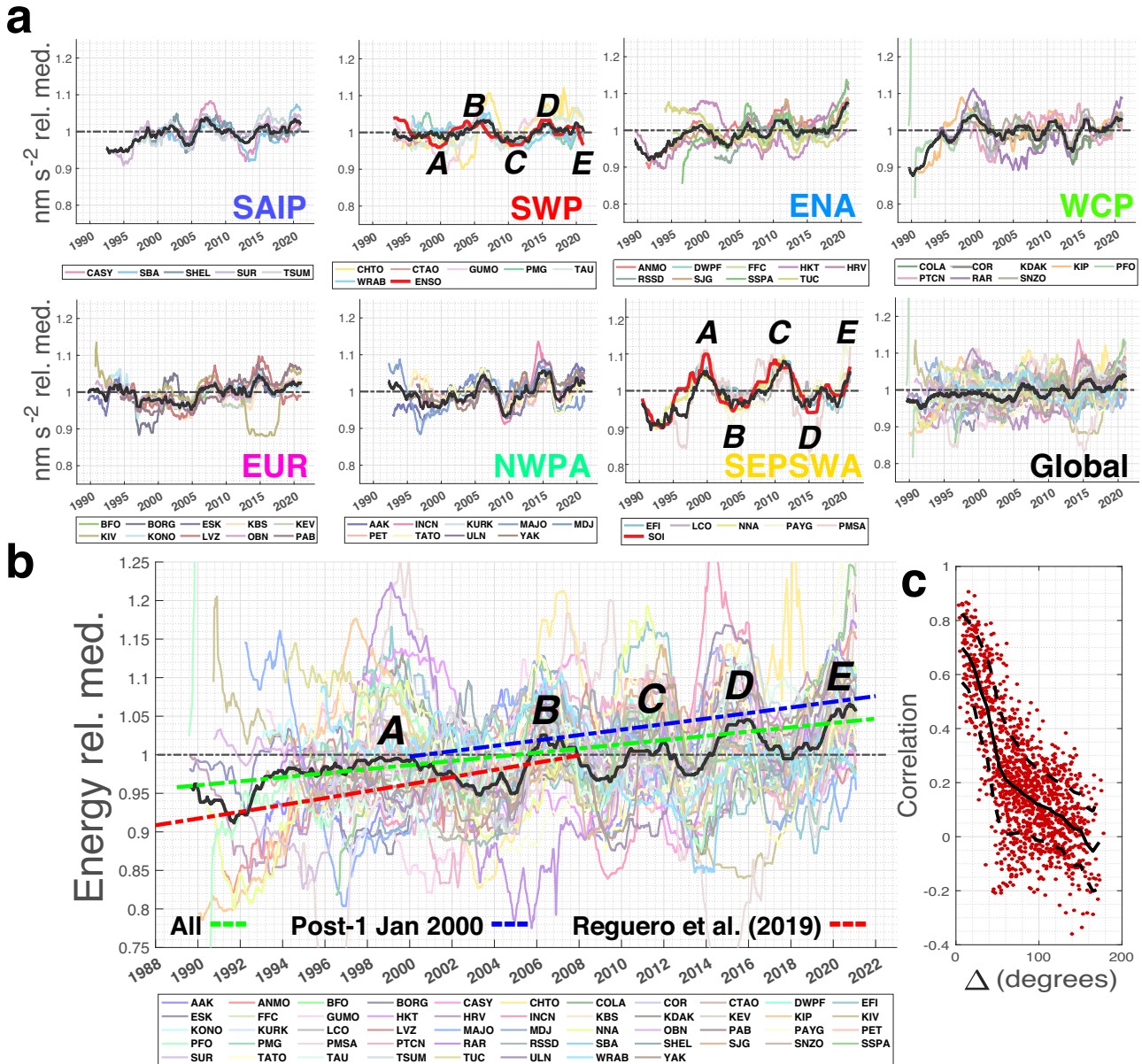

**Fig. 7 | Vertical component acceleration and vertical component seismic energy time series (3-year smoothing), and time series (61-day smoothing) correlations with inter-station angular distance. a** Microseism acceleration histories clustered using the dendrogram of Fig. 5 with three-year moving median data smoothing as in Fig. 1. Figure 6 shows the corresponding 61-day smoothed time series. ENSO and SOI indices, scaled by seismic time series amplitudes, are plotted in red within their associated SWP and SEPSWA clusters, respectively. Black time series show the median of all smoothed data across each cluster. **b** Global seismic energy time series with seasonal harmonics (equation (1)) removed, normalized by respective station medians (Supplementary Fig. 4), and smoothed with a 3-year moving median window. The median across all-time series is shown in black.

Dashed energy trends correspond to 0.27% y⁻¹ (green) and 0.35% y⁻¹ (blue) from this study for the two indicated data periods, and to 0.47% y⁻¹ (red, for 1948–2008, annually compounded) from ref. 14. Trends are normalized to one at 1 January 2005, and ±15% vertical shifts are imposed on the blue and red trends, respectively, for plotting clarity. Global energy excursions labeled A−E corresponding to 3-year moving median ENSO and SOI (Supplementary Fig. 7) excursions as indicated (red) in the SWP and SEPSWA panels in **a**. Time axis tick marks correspond to 1 January of the indicated years. **c** Correlation versus inter-station great-circle distance for demeaned and detrended 61-day median smoothed time series (Fig. 6). Black curves show correlation mean and ±1 standard deviation with 7.5° smoothing.

within the 1-hour segments after linearly detrending each contiguous data segment to avoid broadband discontinuity-induced spectral artifacts[60]. For each subwindow we apply a Hann taper and calculate their average, removing the instrument response to obtain physical units as parameterized by authoritative EarthScope Data Management Center metadata, and returning acceleration PSD estimates in dB relative to 1 (ms⁻²)²/Hz. We estimate microseism band square root integrated power by integrating the hour-long PSD estimates in microseism period bands using the trapezoidal rule and then obtain a time series of the band-limited root power in acceleration units

(ms⁻²). Acceleration spectra are integrated in the frequency domain and squared to obtain seismic energy metrics. Integration to velocity to estimate the velocity squared energy proxy does not have a large effect on proportional amplitude trend statistics but makes metrics more sensitive to lower-frequency signals and produces a secondary effect on station trend ranking (Tables 2, 3).

**Robust trend estimation**

The robust parameter estimates in this study solve for intercept and trend (slope) parameters for the linear function that minimizes the

outlier-resistant $\ell_1$-norm metric

$$\| \mathbf{d} - \mathbf{Gm}\|_1 = \sum_{i=1}^{m} \mathrm{abs}\,(d_i - (\mathbf{Gm})_i) \qquad (6)$$

where $\mathbf{m}$ is the 2-parameter model vector specifying the linear trend intercept and slope, $\mathbf{G}$ is an $n$ by 2 system matrix consisting of a first $n$-length column with each element $G_{i,1} = 1$ and a second $n$-length column $G_{i,2} = t_i$ where $t_i$ is the time of the $i$th data point, and $\mathbf{d}$ is an $n$-element acceleration amplitude or seismic energy time series vector. We do not interpolate across data gaps. The slope and $y$-intercept parameters and their covariance matrix $\mathbf{C}$ are estimated using iteratively re-weighted least squares (IRLS)[61] implemented via the *robustfit* function of MATLAB®. $\mathbf{C}$ is estimated from the convergent least-squares solution of the weighted IRLS equations.

### Dendrogram calculation

Dendrogram associations between the microseism acceleration and climate index time series (Fig. 5) were calculated with MATLAB® from the correlation coefficients between the 54 detrended and demeaned time series shown in Fig. 6. The associated hierarchical cluster tree was generated using single-linkage agglomerative clustering implementing the Ward objective function[62] in which each iterative linking step minimally increases the total within-cluster variance of the time series. The associated separation metric used in the MATLAB® *linkage* function is $D_{ij} = 1 - C_{ij}$ where $C_{ij}$ is the correlation between time series $i$ and $j$.

### Calibration

The Global Seismographic Network (GSN) is jointly operated and maintained by the EarthScope Consortium under funding from the U.S. National Science Foundation and the U.S. Geological Survey. This study utilizes authoritative metadata retrieved for GSN stations from the EarthScope data management system to convert time series counts to physical units. The network incorporates instrumentation configurations that have been increasingly and asynchronously standardized over time. Seismic sensors and data loggers in the network are subject to acceptance testing[63–65] for adherence to manufacturer's specifications (typically ±1% deviation gain and ±5° in phase relative to nominal response). After installation, responses are subject to calibration and consistency tests that have included Earth tides[66], observations of normal modes such as $_0S_0$ from great earthquakes[67], and continuous use and consistency checking of the network for the location and quantification of earthquake sources, seismic tomography, and other applications. The EarthScope DMC also continuously calculates and monitors frequency-dependent quality assurance metrics for GSN and other stations in its ongoing operations[65]. Time-dependent gain reductions[68] in the early GSN noted in a systematic study of $M_W > 6.5$ earthquakes were traced[69] to humidity-related corrosion in feedback electronics in a few percent of Streckheisen STS-1 very broadband sensors. These issues were documented and corrected in the field, and effects on station responses occurred at longer periods than those analyzed here. Anomalous limited (small percentage of total time series) contiguous time periods associated with apparent system malfunctions or incorrect metadata were observed at ten stations after inspection of overlapping hourly PSD integral time series and were excised (Supplementary Fig. 2; Supplementary Table 1.

### Data availability

Global Seismographic Network seismic data in this study are freely and openly available from the EarthScope Consortium using Web Services (https://service.iris.edu; Federation of Digital Seismograph Networks codes II[35] and IU[34]). Bivariate El Niño Southern Oscillation and Southern Oscillation Index (Supplementary Figure 7) time series were downloaded from https://psl.noaa.gov/data/climateindices/list. Spectral estimation files are available at the https://code.usgs.gov/asl/papers/ringler/microseism site maintained by the U.S. Geological Survey. Source data are provided with this (see: Code Availability).

### Code availability

Spectral estimation data files and MATLAB® analysis code used to generate all results and figures are available at the https://code.usgs.gov/asl/papers/ringler/microseism site maintained by the U.S. Geological Survey.

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

## Acknowledgements

The facilities of EarthScope Consortium were used for access to waveforms and related metadata. These services are funded through the Seismological Facility for the Advancement of Geoscience (SAGE) Award of the U.S. National Science Foundation (NSF) under Cooperative Support Agreement EAR-1851048. The Global Seismographic Network (GSN) is a cooperative scientific facility operated jointly by NSF and the United States Geological Survey (USGS). The NSF component is part of the SAGE Facility, operated by EarthScope Consortium under Cooperative Support Agreement EAR-1851048. We made use of the ObsPy Python package and MATLAB® for analysis, including m_map (Pawlowicz, R., 2020. M_Map: a mapping package for MATLAB, version 1.4 m; www.eoas.ubc.ca/~rich/map.html). We offer thanks for helpful comments and suggestions from Janet Carter, Sydney Dybing, and Brian Shiro that significantly improved this manuscript. Any use of trade, firm, and product names is for descriptive purposes only and does not imply endorsement by the U.S. Government.

## Author contributions

R.C.A. led the study, including coding, analysis, coauthor coordination, and writing. A.T.R. calculated PSDs and contributed to analysis and writing. R.E.A. and T.A.L. contributed to conceptualization, analysis, and writing.

## Competing interests

The authors declare no competing interests.
