## [Peer Review File · Nature Communications]

Increasing ocean wave energy observed in Earth's seismic wavefield since the late 20th centuryREVIEWER COMMENTS

Reviewer #1 (Remarks to the Author):

The manuscript presents an analysis on primary microseisms amplitudes for wave periods in the range 14 to 20 s, complementing previous work based on double-frequency microseisms. The particular interest for primary microseism is the expected linear and stable (but frequency dependent) relationship between ocean wave spectra in shallow to intermediate water depths and the primary microseism spectra. As a result, microseism data may be used to infer ocean wave properties in coastal areas from the time scale of a single storm to interannual variability and secular changes.

Here the variability (fig 1) is mostly analyzed in terms trends over a time frame that spans at least 22 years.

I have found the manuscript well written and very interesting, and raising quite a few questions. I thus believe that this paper can be accepted after a minor revision that should help address the following issues.

1) It is not clear that the trends presented here are consistent with published records. One difficulty is the question of calibration of instruments: in situ buoys are infamous for containing spurious trends (e.g. Collins and Jensen 2022), while satellite altimeter require careful calibration and do not discriminate between different periods (e.g. Dodet et al. 2022). Some of the most careful trend analysis was performed by Timmermans (2020) et al. using satellite data, and I believe they are generally consistent with the present results. However, nothing is said about the calibration of seismometers. What is the absolute calibration uncertainty for, say STS-1 or STS-2 instruments?

2) Trends computed over 20-40 years are most often dominated by interannual variability, as demonstrated by Hochet et al. (2023) for the North Atlantic. Probably this issue should be mentioned. In fact, rather than the trends, the interannual patterns (North Atlantic Oscillations, ENSO ...) are probably much more interesting and meaningful: the comparison to Sasaki was particularly interesting when discussing the North Pacific.

Minor issues:

Figure 1: time ticks are oddly placed: 1987, 1993, 1998, 2004, 2009, 2015 ... this is not a regular time interval and it makes detailed comparison to other datasets very challenging (beyond looking at the figure other users may want to actually download the time series). The authors should fix this and clarify if the vertical lines are on January 1st of the year stated below.

there are "???" in references 19 and 57

Reviewer #2 (Remarks to the Author):

Review comment on Aster et al., submitted to Nature Communications

This study presents that the primary microseism amplitudes have increased within 20 years at global scale, using spectral amplitudes of Rayleigh waves at a period of 14–20 s observed in the vertical component of the broadband seismometers. The stations where increases of the primary microseism amplitudes are observed are spatially correlated. The results are very interesting. However, I do not understand how the authors defined quantities used in the text, and also what kind of mechanisms mainly affected the increases of the primary microseism amplitude. I hope that the following comments are helpful when the authors improve the manuscript.

Major comments

1) Definition of quantities

I do not understand how the authors defined quantities used in the text and figures. For example, it is necessary to define the following quantities using equations, such as L1-norm minimizing linear trend and its covariance matrix, proportional rates of microseism amplitude and energy, and absolute rate of amplitude and energy. How did the authors use median values to estimate proportional rate of microseism energy (or amplitude)? Due to the lack of this information, I do not understand the meanings of the quantities. It is also nice if the authors explain with equations how L1-norm is calculated in the `robustfit` function of MATLAB.

2) Mechanisms for the increases of primary microseism amplitudes

Although the quantities used in this study should be defined, it seems that the increases of the primary microseism amplitudes are interesting. However, there are less descriptions on the plausible mechanism that increased the primary microseism amplitude. I would like to read descriptions on the mechanisms with some quantitative estimations. Without such descriptions, it seems that the manuscript is weak for publication in Nature Communications.

The bottom paragraph in Page 6: This description is very interesting. However, when I read this manuscript, I did not understand that the mechanisms stated here really change the primary microseism amplitudes and explain the observed amplitude difference over 20 years. Therefore, I strongly recommend the authors to perform some quantitative estimations. It is a good way if the authors conduct a three-dimensional numerical simulation, maybe at a local scale, in which the Rayleigh waves are generated by the forcing due to ocean swells near coastal areas. If the sea level change or other factors change the generated Rayleigh waves and they are matched with the observed quantities, it is very interesting. Other quantitative estimations with some theoretical approaches are also nice.

3) Sentence structure

It is very hard to read the manuscript, and the sentence structure should be improved thoroughly. For example, the description in Section 1 (1.1 and 1.2) is related to method. The description in Section 2 should be in the result section. The section 3 is too long for conclusion.

Other comments.

The 10th lines from the bottom in Page 2

ear-coastal  near coastal

The 8th line from the bottom in Page 2

“A second and more energetic”

What is “second”?

The 9th line from the top in Page 3

“Meteorological and climate reanalyses and modeling.”

Please rewrite this sentence.

The 14-11th lines from the bottom in Page 4

“We calculate the least-squares...alter general conclusions (Fig. 3)”

This description should be explained using equations.

Also, Fig. 3 is cited before Figs. 1 and 2. Please rearrange the numbering the figures.

Fig 1 caption

applying thee-year  applying three-year

Figs. A5–A8

Why are the velocity amplitudes needed? In the spectra, velocity and acceleration are linked with i -omega, so the tendencies of the amplitude variations between the velocity and acceleration should be the same. It seems that A5–A8 are not needed.

Reviewer #3 (Remarks to the Author):

This manuscript examined long-term temporal variations in microseism energy as an index of near-coastal ocean wave energy and showed a multidecadal increase in microseism and ocean wave energy. The authors used globally distributed seismic stations with long operational periods and showed clear secular increases of microseism energy at most stations and their regionally coherent pattern of the trends. The data are rich, the analysis is generally rigorous, and the results contain interesting findings. Although I do not hesitate to recommend the publication of this work, I feel the presentation can be improved, and some additional figures and analyses can enhance the importance of this work and support the arguments in this manuscript.

Major comments:

1. The differences between the authors' previous work [33] and this study are not very clearly described. This study used the median values as the index of microseisms, which is different from the index (number of extreme microseism events) used by the previous work [33]. I think it would be better to describe how the microseism index in each study reflects the state of ocean waves.
2. The authors argued that “Trends exhibit a high degree of proportionality in that stations with higher

median primary microseism amplitude strongly tend to exhibit greater amplitude and energy increases (Fig. 4)” in Paragraph 2 in Section 2. Although this proportionality is one of the interesting points presented by this work, I do not think that Fig. 4 provides direct evidence of the proportionality between median amplitude and amplitude increasing ratio. This is because a nearly constant median amplitude can produce the trend (proportionality between percent change vs absolute change) shown in Fig. 4. To show the proportionality directly, the authors can show a plot taking the median amplitude in the horizontal axis and the amplitude increasing ratio in the vertical axis. A map showing the median amplitude is also informative.

3. Quantitative comparison with meteorological and oceanographic studies is limited. It would be useful to over-plot time series of long-term wind speed data [20] and (regionally averaged) seismic data of this study and examine correlation coefficients for the time periods over which both data are available. Their regional differences also would be interesting. Such quantitative comparison may enhance the usefulness and importance of microseism-based ocean wave state study including this paper.

Minor comments:

4. Abstract: I do not completely agree with “The primary microseism signal between 20–14 s period is principally composed of seismic Rayleigh waves”. The primary microseisms consist of not only Rayleigh waves but also Love waves, and Love wave energy is as large as Rayleigh wave energy (Nishida et al. 2008, doi:10.1029/2008GL034753).

5. Abstract: It is hard to understand the difference between “the most rapidly proportionately intensifying energy” and “the most rapidly intensifying primary microseism levels on an absolute basis”. This may be due to the lack of the definition of “the proportionately intensifying energy” in the Abstract. I suggest that the authors add a description of it in the Abstract.

6. Paragraph 1, Introduction: “ear-coastal sea floor” may be a typo.

7. Paragraph 2, Introduction: “Meteorological and climate reanalyses and modeling.” does not form a sentence.

8. Paragraph 1, Section 1.2: The authors estimated and removed annual periodic trends by fitting a periodic function to the original time series with the secular trends and then estimated the secular trend from the residual time series. Strictly speaking, however, it is more accurate to fit annual and secular trends simultaneously because there is no modeling error.

9. Fig. 1: The spacing of the horizontal tick labels is not constant.

REVIEWER COMMENTS AND RESPONSES

Reviewer #1 (Remarks to the Author):

The manuscript presents an analysis on primary microseisms amplitudes for wave periods in the range 14 to 20 s, complementing previous work based on double-frequency microseisms. The particular interest for primary microseism is the expected linear and stable (but frequency dependent) relationship between ocean wave spectra in shallow to intermediate water depths and the primary microseism spectra. As a result, microseism data may be used to infer ocean wave properties in coastal areas from the time scale of a single storm to interannual variability and secular changes.

Here the variability (fig 1) is mostly analyzed in terms trends over a time frame that spans at least 22 years.

I have found the manuscript well written and very interesting, and raising quite a few questions. I thus believe that this paper can be accepted after a minor revision that should help address the following issues.

1) It is not clear that the trends presented here are consistent with published records. One difficulty is the question of calibration of instruments: in situ buoys are infamous for containing spurious trends (e.g. Collins and Jensen 2022), while satellite altimeter require careful calibration and do not discriminate between different periods (e.g. Dodet et al. 2022). Some of the most careful trend analysis was performed by Timmermans (2020) et al. using satellite data, and I believe they are generally consistent with the present results. However, nothing is said about the calibration of seismometers. What is the absolute calibration uncertainty for, say STS-1 or STS-2 instruments?

We appreciate the comments regarding buoy data and the referrals to additional references, which we have added in context to the Discussion. We agree regarding the uniqueness of period-dependent information in these data, and this is indeed a promising target for further study.

We added a GSN calibration subsection describing tolerances, calibration practices, and relevant references to the Methods section. The data and metadata receive continuous attention from the seismological community for earthquake and other studies (because these are trend measurements, and much

of the analysis uses historical median normalized data the absolute calibrations are not as important as the stability of station responses here).

We also plotted the Reguero et al [20] global trend in our (new) Fig. 7 to provide graphical comparison to our energy trend estimates (please see points in (2) below). We note consistent El Nino/La Nina trends (Fig. 7a,b) with those noted by Reguero, showing that those (mid-ocean) trends area also clearly reflected in the near-coastal energy environment for these later data as indicated by primary microseism records from the western and southeastern Pacific regions, respectively.

2) Trends computed over 20-40 years are most often dominated by interannual variability, as demonstrated by Hochet et al. (2023) for the North Atlantic. Probably this issue should be mentioned. In fact, rather than the trends, the interannual patterns (North Atlantic Oscillations, ENSO ...) are probably much more interesting and meaningful: the comparison to Sasaki was particularly interesting when discussing the North Pacific.

This set of points (and the above reference) are now amplified in the Discussion and revised figures (see new Figs. 6 and 7, below). The revised manuscript also in this regard incorporates a correlation-based station history clustering (a, below) in response to this comment and to comments from other reviewers. Among other points this clarifies ENSO and SOI index correlations with Southwestern Pacific (cluster SWP) and Southeastern Pacific/Southwestern Atlantic (cluster SEPSWA) regional microseism intensities, respectively. These new figures also demonstrate the imprint of ENSO/SOI excursions in Earth's global median microseism wavefield (El Nino and La Nina events A – E noted in Fig. 7b). The SWP and SEPSWA associations corroborate with the pre-2009 analysis of Reguero et al. (2019; their Fig. 7b) who noted correlations of global wave power with the Niño3 standardized index in which increased ocean wave energy occurs in the western equatorial Pacific and southeastern Pacific during positive and negative excursions, respectively.

Fig. 6 Microseism acceleration time series clustered using the associations of Fig. 5 with seasonal harmonics (equation (1)) removed, normalized by respective station medians (Suppl. Fig. 11) and smoothed with a 61-day moving median window. Fig. 7 shows corresponding 3-year smoothed time series. Black time series show the median of all smoothed time series for each cluster. ENSO and SOI indices (Suppl. Fig. 15) scaled by seismic data are plotted in red within associated SWP and SEPSWA clusters, respectively.

Fig. 7 (a) Microseism acceleration histories grouped using the dendrogram of Fig. 5 with three-year moving median data smoothing as in Fig. 1. Fig. 6 shows the corresponding 61-day smoothed time series. ENSO and SOI indices, scaled by seismic time series amplitudes, are plotted in red within their associated SWP and SEPSWA clusters, respectively. Black time series show the median of all smoothed data across each cluster. (b) Global seismic energy time series with seasonal harmonics (equation (1)) removed, normalized by respective station medians (Suppl. Fig. 11) smoothed with a 3-year moving median window. Black time series is corresponding median across all seismic time series. Dashed energy trends correspond to $0.27\%y^{-1}$ (green) and $0.35\%y^{-1}$ (blue) from this study for the two indicated data periods, and to $0.47\%y^{-1}$ (red, for 1948–2008, annually compounded) from [22]. Trends are normalized to one at 1 January 2005, and $\pm 15\%$ vertical shift axes imposed on the blue and red trends, respectively, for plotting clarity. Global energy excursions labeled A–E corresponding to 3-year moving median ENSO and SOI (Suppl. Fig. 15) excursions as indicated (red) in the SWP and SEPSWA panels in (a). Time axis tick marks correspond to 1 January of indicated years. (c) Correlation versus inter-station great-circle distance for demeaned and detrended 61-day median smoothed time series (Fig. 6). Black curves show correlation mean and ± 1 standard deviation with 7.5° smoothing.

Minor issues:

Figure 1: time ticks are oddly placed: 1987, 1993, 1998, 2004, 2009, 2015 ... this is not a regular time interval and it makes detailed comparison to other datasets very challenging (beyond looking at the figure other users may want to actually download the time series). The authors should fix this and clarify if the vertical lines are on January 1st of the year stated below.

This (date roundoff problem in the plotting program) has been corrected, a corrected figure was uploaded, and the tick mark timing was clarified in the caption as suggested above.

there are "???" in references 19 and 57

This has been corrected (the missing publisher addresses were added in the bibliography file).

Reviewer #2 (Remarks to the Author):

Review comment on Aster et al., submitted to Nature Communications

This study presents that the primary microseism amplitudes have increased within 20 years at global scale, using spectral amplitudes of Rayleigh waves at a period of 14–20 s observed in the vertical component of the broadband seismometers. The stations where increases of the primary microseism amplitudes are observed are spatially correlated. The results are very interesting. However, I do not understand how the authors defined quantities used in the text, and also what kind of mechanisms mainly affected the increases of the primary microseism amplitude. I hope that the following comments are helpful when the authors improve the manuscript.

Thank you for these comments. The revised manuscript clarifies our methodology and the quantities calculated in the paper, as well as including amplified discussion regarding the details of the primary microseism source mechanism.

Major comments

1) Definition of quantities

I do not understand how the authors defined quantities used in the text and figures. For example, it is necessary to define the following quantities using equations, such as L1-norm minimizing linear trend and its covariance matrix, proportional rates of microseism

amplitude and energy, and absolute rate of amplitude and energy. How did the authors use median values to estimate proportional rate of microseism energy (or amplitude)? Due to the lack of this information, I do not understand the meanings of the quantities. It is also nice if the authors explain with equations how L1-norm is calculated in the robustfit function of MATLAB.

We added (Methods) information on the robust trend fitting theory and methodology employed here, including a reference in which the iterative re-weighted least squares (IRLS) procedure is described. We have also noted how the covariance matrix is estimated from the reweighted system of least-squares equations at the convergence of this iterative algorithm. Proportional trends were calculated by normalizing the trends by the historical median values observed at each station (equations 2 and 3), and we incorporating a naming convention for the various time series referred to in the manuscript to make these quantities clearer for readers.

2) Mechanisms for the increases of primary microseism amplitudes

Although the quantities used in this study should be defined, it seems that the increases of the primary microseism amplitudes are interesting. However, there are less descriptions on the plausible mechanism that increased the primary microseism amplitude. I would like to read descriptions on the mechanisms with some quantitative estimations. Without such descriptions, it seems that the manuscript is weak for publication in Nature Communications.

Thank you for inviting further elaboration on this central topic. Our principal aim in this (shorter format) paper is to report and quantitatively document the increasing energy in Earth's background seismic and causative ocean wavefield in a temporally and geographically extensive observational study using robust estimation. As we describe in the expanded Discussion, the proposed mechanism for widespread and systematic increase in primary microseism amplitudes (and the only one that we ascribe to be credible) is a systematic increase in the amplitude of the causative tractions on the seafloor at periods between about 20 and 14 s in at depths of less than several hundred meters (we added the relevant equations 4 and 5 in this). The primary microseism mechanism is sufficiently well understood to contend that these time series are records of spatiotemporally integrated wave-induced ocean bottom forces across broad regions of the Earth (e.g., Fig. 7c). In quantitatively associating seismic energy increase with near-coastal wave energy increase, we appeal to linearity as elaborated in the Results section. We have additionally augmented the relevant references and elaborated more clearly outstanding challenges associated with full modeling of primary microseism.

The bottom paragraph in Page 6: This description is very interesting. However, when I read this manuscript, I did not understand that the mechanisms stated here really change the primary microseism amplitudes and explain the observed amplitude difference over 20 years. Therefore, I strongly recommend the authors to perform some quantitative estimations. It is a good way if the authors conduct a three-dimensional numerical simulation, maybe at a local scale, in which the Rayleigh waves are generated by the forcing due to ocean swells near coastal areas. If the sea level change or other factors change the generated Rayleigh waves and they are matched with the observed quantities, it is very interesting. Other quantitative estimations with some theoretical approaches are also nice.

We appreciate this comment and respectfully assert that the 3-D modeling recommendation in its fullest form extends beyond the scope of this paper, which provides solid motivation for further advancement in this as a well warranted and aspirational goal. At present this is a challenging undertaking even in lower-dimensional simulations. For example, Gualtieri et al. (2019), using the theoretical approach of Hasselmann (1963) and Arduin et al. (2015, 2018) were able to demonstrate general agreement between modeling and observations for the primary mechanism using a (linearized) 1-D Fourier method implementing wave and bathymetric spectra in combination with WAVEWATCH III estimates, but noted further complexities that need focused attention including the nonlinear influences of steeper slopes, partitioning of energy into multimodal surface and love waves, and 3-D (including wave directionality and small-scale bathymetry) coupling effects that necessitated the introduction of an empirical fitting parameter in their simulations. Empirically, this clarifies the broad source regions at play and highlights the extensive spatial integration involved (consistent with that noted by Gualtieri, 2019) in its correlations between El Nino and SW Pacific intensity, and between La Nina/SOI and SE Pacific/SW Atlantic intensity (Fig. 7a), as well as in the extensive spatial correlation lengths between station time series depicted in Fig. 7c (which confirms the findings of Gualtieri (2019) the primary microseism signal at a particular station can be sensitive to sections of coast at ranges of up to 1000's of km).

More broadly, our principal conclusion that the causative effect of the observed trends is an increase in these seafloor forces over the past 3--4 decades due to increasing wave energy performing elastic work on the seafloor is, we contend, very well supported in this paper and in associated references. Because the changes are (proportionately) small relative to the median or average excitation levels, we invoke the reasonable assumption of quasi-linearity in our analysis to associate seismic and wave energy rates of increase (as described in the revised manuscript).

Equation 5 in the revised manuscript notes the dynamic pressure period/depth relationship for linearized ocean swell to illuminate the essential relationship between wave height and the (ocean floor) seismic source for the primary microseism. The reviewer's perhaps broader point regarding sea level changes (if we are interpreting this correctly) possibly affecting this signal is an intriguing one that we previously investigated by probing our time series for evidence of tidal period modulations in primary microseism amplitudes using Lomb-Scargale (aperiodic sampling) spectral estimation. However, this only resolved the annual and annual harmonic periodicities in microseism amplitude and intensity associated with the seasonal cycle (the large $H(t)$ annual signal components described in the manuscript). Since the process is excited by waves at ocean swell periods (with wavelengths of ~300 – 600 m) and the sensitivity footprint is widespread (Fig. 7c) tidal-scale sea level changes do not appear so far to have a discernable signature.

3) Sentence structure

It is very hard to read the manuscript, and the sentence structure should be improved thoroughly. For example, the description in Section 1 (1.1 and 1.2) is related to method. The description in Section 2 should be in the result section. The section 3 is too long for conclusion.

The manuscript has been extensively rewritten reformatted in section organization, scientific units, and otherwise for Nature Communications (per <https://www.nature.com/ncomms/submit/article>).

We have also edited the manuscript throughout and received confidential internal reviews from colleagues within our institutions to assure general textural and mathematical readability, linkages with figures, figure ordering, and otherwise address the reviewer's general readability comment in mind. We explicitly specified and cross-referenced relevant equations and implemented a time series naming convention throughout the manuscript to further improve its accessibility for a wide audience.

Other comments.

The 10th lines from the bottom in Page 2
ear-coastal  near coastal

Corrected

The 8th line from the bottom in Page 2

“A second and more energetic”
What is “second”?

Clarified -- this was in reference to the second (secondary) microseism source mechanism.

The 9th line from the top in Page 3
“Meteorological and climate reanalyses and modeling.”
Please rewrite this sentence.

This sentence fragment has been removed.

The 14-11th lines from the bottom in Page 4
“We calculate the least-squares...alter general conclusions (Fig. 3)”
This description should be explained using equations.

Done. We also introduced a naming convention, with the functions $A(t)$, $E(t)$ corresponding to the key time series, and the stationary seasonal harmonic projections $H(t)$. The relevant equation for $H(t)$ is noted (equation 1).

Also, Fig. 3 is cited before Figs. 1 and 2. Please rearrange the numbering the figures.

We re-checked and corrected the figure order in the revised manuscript

Fig 1 caption
applying thee-year  applying three-year

Corrected.

Figs. A5–A8
Why are the velocity amplitudes needed? In the spectra, velocity and acceleration are linked with i - ω , so the tendencies of the amplitude variations between the velocity and acceleration should be the same. It seems that A5–A8 are not needed.

These signals have a modest (~6 s) bandwidth. Thus, conversion from acceleration to velocity has some effect on corresponding metrics derived from the integrated PSDs (e.g., weighting longer periods as $1/\omega$ in the spectra). The spectra are fairly narrow-band (6-s) and smooth (varying by a few dB) and the reviewer is correct in suspecting that relevant differences between these integrals are modest in the context of this study (e.g., compare Figs. 1 and Supp. Fig. 10 in the revised manuscript, the global acceleration history and global energy history in Fig. 7a,b, or the spectra for GEOSCOPE network stations in

Gualtieri (2019). We agree that the seismic velocity amplitude plots and table in the original manuscript were superfluous and they have been removed along with associated references to velocity series throughout the paper.

Reviewer #3 (Remarks to the Author):

This manuscript examined long-term temporal variations in microseism energy as an index of near-coastal ocean wave energy and showed a multidecadal increase in microseism and ocean wave energy. The authors used globally distributed seismic stations with long operational periods and showed clear secular increases of microseism energy at most stations and their regionally coherent pattern of the trends. The data are rich, the analysis is generally rigorous, and the results contain interesting findings. Although I do not hesitate to recommend the publication of this work, I feel the presentation can be improved, and some additional figures and analyses can enhance the importance of this work and support the arguments in this manuscript.

Major comments:

1. The differences between the authors' previous work [33] and this study are not very clearly described. This study used the median values as the index of microseisms, which is different from the index (number of extreme microseism events) used by the previous work [33]. I think it would be better to describe how the microseism index in each study reflects the state of ocean waves.

We clarified the relationship between the previous work and these studies within the Introduction section of the revised manuscript. The earlier (Aster et al.) index that the reviewer refers to examined extreme outlier statistics of seismic data (e.g., extreme wave events in the record observed over short [a few hours] time intervals) while this study presents observations of characteristic (median) trends in amplitude and energy over a more recent and well-calibrated most recent 3-4 decades of the global seismic record, smoothed over 2-month and 3-year time scales. IN addition, the data examined here from the last 15 years or so were not available at the time of publication for the previous study and show much more robust evidence of global secular change than the shorter time series. We agree with the reviewer that the data and methodologies in this paper indeed reveal a rich trove of multi-scale spatiotemporal statistical information relevant to evolving ocean wave state and hope that the study will inspire future work.

2. The authors argued that "Trends exhibit a high degree of proportionality in that stations with higher median primary microseism amplitude strongly tend to exhibit greater amplitude and energy increases (Fig. 4)" in Paragraph 2 in Section 2. Although this proportionality is one of the interesting points presented by this work, I do not think

that Fig. 4 provides direct evidence of the proportionality between median amplitude and amplitude increasing ratio. This is because a nearly constant median amplitude can produce the trend (proportionality between percent change vs absolute change) shown in Fig. 4. To show the proportionality directly, the authors can show a plot taking the median amplitude in the horizontal axis and the amplitude increasing ratio in the vertical axis.

Thank you for this comment -- we agree that the inherent correlation in plotting two dependent variables against each other was confusing in this figure. As suggested by the reviewer, we replaced the original figure with a straightforward and we suggest more informative plot of absolute trend versus median proportionality as suggested (new Fig. 4; below). This plot reinforces the dominance of positive trends, shows where various stations lie (with the various dotted-line linear trends) and further displays trend variations with general region. We have also modified the text accordingly in the discussion of this proportionality effect, which is evident but modest for some of the regional arms as indicated by color (correlation of 0.213) with proportionality being most evident for the North American and European Atlantic regions (red, blue, respectively) and South Pacific (green and teal) regions/stations.

Fig. 4 Vertical-component acceleration amplitude trends R_A calculated with seasonal harmonics subtracted (Figs. 1a, 2a; Table 1) versus historical station median with 3σ confidence intervals. Correlation coefficient is 0.213 and 13 stations exhibit positive trends at 3σ significance that are greater in absolute value than at the most negative station (HKT; Hockley, Texas). Colors reflect geographic groups defined in Fig. 1. Dotted lines indicate representative percentage amplitude changes P_A relative to the historical station median (Figs. 1b, 2b, 11; Table 1).

A map showing the median amplitude is also informative.

We added a map figure of median amplitudes to the manuscript (Supp. Fig. 11; below).

Fig. 11 (Supplemental) Median primary microseism vertical acceleration amplitudes for the operational history of each station through 1 August 2022.

3. Quantitative comparison with meteorological and oceanographic studies is limited. It would be useful to over-plot time series of long-term wind speed data [20] and (regionally averaged) seismic data of this study and examine correlation coefficients for the time periods over which both data are available. Their regional differences also would be interesting. Such quantitative comparison may enhance the usefulness and importance of microseism-based ocean wave state study including this paper.

We significantly fleshed out these critical points of connection in the revised manuscript and graphically display these associations in time series format in the new Figs. 5 and 7 (below). Specifically, the revised manuscript incorporates a dendrogram analysis of station histories that notes the revealed geographically associations (Fig 5b). We associated time series correlations with the Southern Oscillation and Bivariate ENSO indices, which are nearly anticorrelated (<https://psl.noaa.gov/data/climateindices/list>) and note ENSO influence in the regional and global data.

The Reguero et al. (initial manuscript reference [20]) analysis is a solid and well-known touchstone for independent remote sensing estimates of global wave energy but ends in 2008 (e.g., their Figs. 2 and 3) and thus does not overlap with notable regional and global features shown in the (new) Fig. 7. We have included

additional discussion regarding regional and global sensitivity in the seismic data to ENSO excursions.

Fig. 7 in the revised manuscript now compares both our linear trends from this study with the overall (1948-2008) trend from Reguero et al. [20]. Figs. 5 and 7 indicate that southwestern Pacific stations (WRAB, TAU, PMG, CTAO, GUMO, and CHTO; cluster SWP) cluster with ENSO (\sim -SOI), and Eastern Pacific/South Atlantic/Antarctic stations (EFI, LCO, NNA, PAYG, PMSA; cluster SEPSWA) correlate and cluster with SOI (\sim the additive inverse of ENSO) across the multi-decadal observational period. This result is consistent as noted in the new manuscript with the conclusions displayed in and drawn from Fig. 7 by the Reguero et al. for 1948—2010. We tested for correlations with the other indices retrieved from the above NOAA site and did not identify significant associations at this time scale and temporal smoothing window.

Correlation analysis included in this revised manuscript further clarified the long-range extent of primary microseism geographic averaging, with interstation correlations between time series showing significance to station separations beyond 50° (Fig. 7 c) thus further demonstrating the extensive geographic averaging within the primary microseism signal. Importantly, this shows that the primary microseism signal as processed here (Figs. 6, 7a) via the use of 2-month temporal smoothing and normalization by local historical median levels provides a consistent proxy for long-range ocean wave intensity across the substantial sectors of the Earth identified in our clustering algorithm and as evidenced by general time series consistency from station to station across the clustered regions identified in this study and supported by the correlative associations with the ENSO and SOI indices.

Fig. 5 (a) Microseism acceleration station clustering derived from correlation (Fig. 6, Suppl. Fig. 14) of detrended 61-day moving median microseism acceleration amplitude time series with seasonal harmonics (equation (1)) and secular trends (Fig. 1; Table 1) removed. D denotes the Ward dissimilarity metric [85]. (b) Dendrogram classification of station groups corresponding to (a): SWP: Southwest Pacific; SAIP: South Atlantic, Indian, Pacific; SEPSWA: Southeast Pacific and Southwest Atlantic; WCP: Western and Central Pacific; ENA: Eastern North America; NWPEA: Northwest Pacific and east Asia; EUR: Europe and Southwest Asia. ENSO and SOI indicate dendrogram correlation-based associations for equivalently smoothed El Niño and Southern Oscillation index time series (Suppl. Fig. 15). Regions of ocean bathymetry with depths of less than 300 m and candidate primary microseism source zones (equation (5)) correspond to the color bar. Transparent geographic boundaries and colors correspond to the general regional associations (Fig. 1, Suppl. Fig. 10).

Fig. 7 a) Microseism acceleration histories grouped using the dendrogram of Fig. 5 with three-year moving median data smoothing as in Fig. 1. Fig. 6 shows the corresponding 61-day smoothed time series. ENSO and SOI indices, scaled by seismic time series amplitudes, are plotted in red within their associated SWP and SEPSWA clusters, respectively. Black time series show the median of all smoothed data across each cluster. b) Global seismic energy time series with seasonal harmonics (equation (1)) removed, normalized by respective station medians (Supp. Fig. 11) smoothed with a 3-year moving median window. Black time series is corresponding median across all seismic time series. Dashed energy trends correspond to $0.27\%y^{-1}$ (green) and $0.35\%y^{-1}$ (blue) from this study for the two indicated data periods, and to $0.47\%y^{-1}$ (red, for 1948–2008, annually compounded) from [26]. Trends are normalized to one at 2005 and a $\pm 15\%$ vertical shift is imposed on the blue and red trends, respectively, for plotting clarity. Global energy excursions labeled A–E corresponding to 3-year moving median ENSO and SOI excursions as indicated in the SWP and SEPSWA panels in (a) (Supp. Fig. 15) Time axis tick marks correspond to 1 January of indicated years. c) Correlation versus inter-station great-circle distance for demeaned and detrended 61-day median smoothed time series (Fig. 6). Black curves show correlation mean and ± 1 standard deviation with 7.5° smoothing.

Minor comments:

4. Abstract: I do not completely agree with “The primary microseism signal between 20–14 s period is principally composed of seismic Rayleigh waves”. The primary microseisms consist of not only Rayleigh waves but also Love waves, and Love wave energy is as large as Rayleigh wave energy (Nishida et al. 2008, doi:10.1029/2008GL034753).

The reviewer’s point is correct -- we clarified in the abstract and text that this study uses *vertical-component* data that are insensitive to Love waves (and to Rayleigh wave propagation direction as well) and are dominated by fundamental mode Rayleigh waves (e.g., Schimmel et al., 2011; DOI: 10.1029/2011GC003661). We also added an introductory sentence to better recognize the foundational Nishida et al. and other papers that address the multiplicity wave types within the microseism wavefield.

5. Abstract: It is hard to understand the difference between “the most rapidly proportionately intensifying energy” and “the most rapidly intensifying primary microseism levels on an absolute basis”. This may be due to the lack of the definition of “the proportionately intensifying energy” in the Abstract. I suggest that the authors add a description of it in the Abstract.

We have clarified in the abstract and elsewhere that reported proportionate trends are relative to long-term station median values and added relevant equations and symbolic definitions (equations 2 and 3) within the text, including noting explicitly in the abstract that reported percentage trends at each station are relative to long-term station median values.

6. Paragraph 1, Introduction: “ear-coastal sea floor” may be a typo.

Corrected.

7. Paragraph 2, Introduction: “Meteorological and climate reanalyses and modeling.” does not form a sentence.

This sentence fragment has been deleted.

8. Paragraph 1, Section 1.2: The authors estimated and removed annual periodic trends by fitting a periodic function to the original time series with the secular trends and then

estimated the secular trend from the residual time series. Strictly speaking, however, it is more accurate to fit annual and secular trends simultaneously because there is no modeling error.

Because the trend and y-intercept and L-1 norm fitting functions are orthogonal to the trigonometric basis functions that characterize stationary annual and harmonic trends the two approaches yield effectively identical results. We also verified that four Fourier terms were sufficient to remove stationary seasonality by examining the complex coefficient amplitudes for each stations, The complex harmonic coefficient amplitudes for acceleration amplitude data sorted by latitude are shown in the (not-included in manuscript) figure below, which also shows the N-S hemispherical phasing and other interesting details of this signal. We also examined spectral lines within the Lomb-Scargale spectrum of the daily-sampled multidecadal time series to search for any additional periodicities (e.g., tidal) but these were not apparent.

9. Fig. 1: The spacing of the horizontal tick labels is not constant.

This (labeling numerical roundoff) issue has been corrected in the new Figs. 1 and Supp. 10.

REVIEWERS' COMMENTS

Reviewer #1 (Remarks to the Author):

The authors have made a thorough effort to consider all previous comments, with some interesting new figures and updated text, including now a mention of interannual variability in the abstract. Also it is very important that defined in detail the quantities that are used.

I believe the paper can be accepted as is.

Obviously the new figures lead to a few more questions (like: why is the KIV curve behaving the way it is ? Could there be an issue with instrument response there? ... I've delayed my review because I wanted to check on this, using microseism spectra computed by IPGP - these are available as part of a European Space Agency project on climate time series - but could not find the time)

Thank you again for an in depth revision.

Reviewer #2 (Remarks to the Author):

Comments to Aster et al., submitted to Nature Communications

I reviewed the previous version of the manuscript, and I am almost satisfied with the current version of the manuscript. The authors did a great job. Thank you very much. It is nice that the authors address the following points before publication of this study.

Main points

7th-8th lines in the second paragraph in Page 4

that predicted  that are predicted

Page 7

The authors should add more explanations for making the dendrogram in Fig. 5 using the correlation coefficient in Fig. 14. It is necessary to explain how the clustering is performed with citing some references.

7th line from the bottom in the second paragraph in Page 7

subsequent excursions, perhaps due to lower data

subsequent excursions, is perhaps due to lower data

Please check this part again.

3rd line in the third paragraph in Page 7

a slope change near 50°

a slope change near a distance of 50°

5th line from the top in Page 8

Please finish the final part of “station within the”.

4th line in the second paragraph in Page 9

this signals reflects

these signals reflect

Please check this part again.

Equation (6)

Please define the column and row sizes of matrices of m and G , and their elements precisely. What are “ones” in “a column of ones”?

Equation (4) and the caption of Fig. 1

Both contents include the character of λ (wavelength and longitude), which confused readers. It is better to change either.

Caption of Fig. 5

The two sentences after “Regions of ocean bathymetry with depths” should be moved for the explanations for panel (a).

Reviewer #3 (Remarks to the Author):

The authors have adequately addressed all comments on the previous manuscript, and the manuscript has been greatly improved. The correlation between primary microseisms and the ENSO indices, added in the revision, is fascinating. I believe this research will have a major impact on a wide range of research fields. I would like to add just minor comments, which can be addressed by the authors before the publication of the manuscript.

Minor comments:

1. Fig. 7b indicates that both positive and negative ENSO phases tend to increase the global average energy of primary microseisms (may be due to teleconnection). However, the SWP and SEPWA regions show correlations more directly, such as microseism energy increases in positive ENSO phases and decreases in negative ENSO phases. The difference may be confusing. It would be better to clarify the difference in the near and distant influences of ENSO. The authors can display the average microseism time series for the area excluding SWP and SEPWA in addition to the global average and discuss the near and distance ENSO influences.
2. The 23th line from the top in Page 3: ... has not been been ...
3. The 4th line from the top in Page: The reference for the ENSO and SOI indices is necessary here.
4. The 4th line from the top in Page8: station within the ?
5. Figure 4: The unit of the horizontal axis should be nm/s^2 .
6. Figure 6: It is very hard to see the correlation between microseisms time series and ENSO/SOI indices. Improvements are required, such as stretching each panel horizontally.

Reply to REVIEWERS' COMMENTS (rev 2)
Increasing near-coastal ocean wave energy observed in Earth's seismic wavefield since the late 20th century

R. Aster, R. Ringler, R. Anthony

Please note that we have interspersed the authors' responses in bold here.

Reviewer #1 (Remarks to the Author):

The authors have made a thorough effort to consider all previous comments, with some interesting new figures and updated text, including now a mention of interannual variability in the abstract. Also it is very important that defined in detail the quantities that are used.

I believe the paper can be accepted as is.

Obviously the new figures lead to a few more questions (like: why is the KIV curve behaving the way it is? Could there be an issue with instrument response there? ... I've delayed my review because I wanted to check on this, using microseism spectra computed by IPGP - these are available as part of a European Space Agency project on climate time series - but could not find the time)

Thank you for your thorough re-review of the paper. There are indeed many other features of these types of data that warrant additional study, and it's good that the ESA groups are analyzing microseism spectra in their efforts as well.

Reviewer #2 (Remarks to the Author):

Comments to Aster et al., submitted to Nature Communications

I reviewed the previous version of the manuscript, and I am almost satisfied with the current version of the manuscript. The authors did a great job. Thank you very much. It is nice that the authors address the following points before publication of this study.

Main points

7th-8th lines in the second paragraph in Page 4
that predicted  that are predicted

Corrected.

Page 7

The authors should add more explanations for making the dendrogram in Fig. 5 using the correlation coefficient in Fig. 14. It is necessary to explain how the clustering is performed with citing some references.

We added a Methods subsection describing how the dendrogram was generated that includes appropriate seminal reference (Ward, 1963) and specific mention of the MATLAB linkage function and dissimilarity measure used.

7th line from the bottom in the second paragraph in Page 7
subsequent excursions, perhaps due to lower data
subsequent excursions, is perhaps due to lower data
Please check this part again.

This point has been split into two sentences to improve readability.

3rd line in the third paragraph in Page 7
a slope change near 50°
a slope change near a distance of 50°

Corrected.

5th line from the top in Page 8
Please finish the final part of "station within the".

This sentence fragment has been corrected. The associated sentence now reads:

This correlation length scale indicates that primary microseism amplitudes and energies provide consistent proxies for large spatial and temporal scale ocean wave variability.

4th line in the second paragraph in Page 9
this signals reflects

these signals reflect
Please check this part again.

Corrected to: ... indicate that these signals reflect...

Equation (6)

Please define the column and row sizes of matrices of m and G , and their elements precisely. What are “ones” in “a column of ones”?

These points are now clarified in the associated Methods subsection (“ones” means all of the elements in the column are equal to 1).

Equation (4) and the caption of Fig. 1

Both contents include the character of λ (wavelength and longitude), which confused readers. It is better to change either.

We revised the caption to use the script lower case “ l ” symbol to indicate longitude.

Caption of Fig. 5

The two sentences after “Regions of ocean bathymetry with depths” should be moved for the explanations for panel (a).

Done.

Reviewer #3 (Remarks to the Author):

The authors have adequately addressed all comments on the previous manuscript, and the manuscript has been greatly improved. The correlation between primary microseisms and the ENSO indices, added in the revision, is fascinating. I believe this research will have a major impact on a wide range of research fields. I would like to add just minor comments, which can be addressed by the authors before the publication of the manuscript.

Minor comments:

1. Fig. 7b indicates that both positive and negative ENSO phases tend to increase the global average energy of primary microseisms (may be due to teleconnection). However, the SWP and SEPWA regions show correlations more directly, such as microseism energy increases in positive ENSO phases and decreases in negative ENSO phases. The difference may be confusing. It

would be better to clarify the difference in the near and distant influences of ENSO. The authors can display the average microseism time series for the area excluding SWP and SEPWA in addition to the global average and discuss the near and distance ENSO influences.

This is a very interesting point that we agree is worthy of further elaboration. We added a supplemental figure (Supp. Figure 8) that is constructed identically to Figure 7b but excludes the two station clusters (SWP and SEPWA) and associated 11 stations that have the strongest ENSO and SOI correlations. This figure demonstrates that the ENSO-correlated global microseism level peaks remain visible even outside of the most strongly correlated regions (consistent with Figure 7c). The extensive geographic signature of ENSO cycles and other time series features reflects some combination of Rayleigh wave propagation, long-distance swell teleconnection, and meteorological correlation in trans-basin storminess.

2. The 23th line from the top in Page 3: ... has not been been ...

Corrected.

3. The 4th line from the top in Page: The reference for the ENSO and SOI indices is necessary here.

We have now cited the seminal Smith et al. (2000) and Ropelewski and Halpert (1987) references with respect to these indices and in association with the associated ENSO indices supplementary figure.

4. The 4th line from the top in Page8: station within the ?

This sentence fragment has been corrected. The associated sentence now reads:

This correlation length scale indicates that primary microseism amplitudes and energies provide consistent proxies for long-range ocean wave variability.

5. Figure 4: The unit of the horizontal axis should be nm/s^2 .

Corrected.

6. Figure 6: It is very hard to see the correlation between microseisms time series and ENSO/SOI indices. Improvements are required, such as stretching each panel horizontally.

The widths of the referenced plots in Fig. 6 have been doubled to address this issue.